# Systemic brain tumor delivery of synthetic protein nanoparticles for glioblastoma therapy

Jason V. Gregory [1,2,8], Padma Kadiyala[3,4,8], Robert Doherty[3,4], Melissa Cadena[1,5], Samer Habeel[1,5], Erkki Ruoslahti[6,7], Pedro R. Lowenstein[1,3,4], Maria G. Castro [1,3,4✉] & Joerg Lahann [1,2,5✉]

Glioblastoma (GBM), the most aggressive form of brain cancer, has witnessed very little clinical progress over the last decades, in part, due to the absence of effective drug delivery strategies. Intravenous injection is the least invasive drug delivery route to the brain, but has been severely limited by the blood-brain barrier (BBB). Inspired by the capacity of natural proteins and viral particulates to cross the BBB, we engineered a synthetic protein nano-particle (SPNP) based on polymerized human serum albumin (HSA) equipped with the cell-penetrating peptide iRGD. SPNPs containing siRNA against Signal Transducer and Activation of Transcription 3 factor (STAT3i) result in in vitro and in vivo downregulation of STAT3, a central hub associated with GBM progression. When combined with the standard of care, ionized radiation, STAT3i SPNPs result in tumor regression and long-term survival in 87.5% of GBM-bearing mice and prime the immune system to develop anti-GBM immunological memory.

[1] Biointerfaces Institute, University of Michigan, 2800 Plymouth Road, Ann Arbor, MI 48109, USA. [2] Chemical Engineering, University of Michigan, 2800 Plymouth Road, Ann Arbor, MI 48109, USA. [3] Department of Neurosurgery, University of Michigan Medical School, 1500 E. Medical Center Drive SPC 5338, Ann Arbor, MI 48109, USA. [4] Department of Cell and Developmental Biology, University of Michigan Medical School, 109 Zina Pitcher Place, Ann Arbor, MI 48109, USA. [5] Biomedical Engineering, University of Michigan, 2200 Bonisteel Blvd, Ann Arbor, MI 48109, USA. [6] Cancer Research Center, Sanford Burnham Prebys Medical Discovery Institute, 10901 North Torrey Pines Road, La Jolla, CA 92037, USA. [7] Center for Nanomedicine and Department of Cell, Molecular and Developmental Biology, Building 235, University of California, Santa Barbara, Santa Barbara, CA 93106, USA. [8] These authors contributed equally: Jason V. Gregory, Padma Kadiyala. ✉email: mariacas@med.umich.edu; lahann@umich.edu

Glioblastoma (GBM) is the most prevalent and aggressive form of brain cancer, currently accounting for ~47% of diagnosed brain cancers[1]. GBM is characterized by its high invasiveness, poor clinical prognosis, high mortality rates, and frequent recurrence[2]. Current therapeutic approaches include focal radiotherapy, and adjuvant chemotherapeutics in combination with surgical resection. However, due to the delicate anatomical structure of the brain and the highly invasive nature of glioma cells, complete surgical resection is rarely achieved[3]. Residual tumor cells infiltrate the surrounding brain tissue and are protected by the blood-brain barrier (BBB), rendering them unresponsive to conventional chemotherapeutic drugs[4]. Despite recent surgical and chemotherapeutic advances, the median survival (MS) for patients diagnosed with GBM remains just 12–15 months with a 5-year survival rate of 5%. The development of alternative and targeted delivery approaches to effectively treat GBM remains one of the most pressing challenges in cancer therapy[5].

A growing body of evidence suggests that the signal transducer and activator of transcription 3 (STAT3) pathway is involved in multiple signaling pathways related to tumor progression and evasion of the immune system[6–8]. Multiple growth factors and cytokines frequently overexpressed in cancer, such as EGF, FGF, and IL-6, activate STAT3 via tyrosine phosphorylation[8–10]. Activated STAT3 (pSTAT3) translocates to the nucleus and participates in the transcription of genes that inhibit apoptosis, and promote tumor cell proliferation and metastasis. Histopathological analysis of brain tumors demonstrated STAT3 to be overexpressed in patients with grade III astrocytomas and grade IV GBMs; increased STAT3 levels are negatively associated with MS in these patients[7]. In previous studies, the STAT3 inhibitor CPA-7 induced tumor cell death in GL26 mouse GBM and HF2303 human GBM stem cells. However, a positive therapeutic effect was only observed in peripheral tumors, but not in intracranial tumors, pointing towards the inability of CPA-7, like many small molecule therapeutics and biomolecules, to cross the protective BBB and enter the brain tumor compartment[6].

A wide range of nanoparticles (NPs) have been developed to deliver chemotherapeutic drugs, such as docetaxel[11–13], paclitaxel[14–16], doxorubicin[17] or other small molecule chemotherapeutics[18–22]; or, to leverage antibodies[23,24], RNA[25–28], or peptides[29] in an attempt to enhance GBM therapy. Despite these grand efforts, research conducted over the past decades has made only marginal advances with no real promise of a clinical path towards a viable curative treatment. In general, these nanocarriers share a number of common characteristics, e.g., (i) they are made of synthetic materials, (ii) they tend to accumulate and persist in liver and spleen causing severe side effects, and (iii) they are incapable of passing the BBB. In contrast, natural evolution has resulted in proteins and viral particulates that can target to and transport through the BBB[30]. Inspired by the unique capabilities of biological NPs, we engineered a GBM-targeting synthetic protein nanoparticle (SPNP) comprised of polymerized human serum albumin (HSA) and oligo(ethylene glycol) (OEG), loaded with the cell-penetrating peptide iRGD[31–33] as well as STAT3i. The choice of HSA as the major matrix component was motivated by its rapid and well-understood clearance mechanisms, its demonstrated clinical relevance, and its exquisite biochemical compatibility with both, therapeutic agents and homing peptides. In addition, albumin-based nanocarriers have been shown to engage cell-surface receptors, such as SPARC[34] and gp60[35], that are overexpressed on glioma cells and tumor vessel endothelium[36–38].

Here we demonstrate the effective systemic delivery of albumin-based SPNPs to aggressive intracranial GBM tumors. The incorporation of the tumor-targeting, tissue-penetrating peptide, iRGD, results in an ability of the SPNPs to penetrate the highly selective BBB and distribute throughout the tumor volume, efficiently delivering siRNA against STAT3 without the use of invasive surgical procedures. When combined with current standard of care, focused radiotherapy, we show the combined therapy to be most effective with 87.5% of mice reaching long-term survival timepoints. When rechallenged with a second tumor in the contralateral hemisphere, these same mice all reach a second long-term survival timepoint in the absence of additional therapeutic intervention. Together, these results suggest that SPNPs are an effective vehicle for the targeted delivery of encapsulated biological therapeutics. Moreover, their use in delivering STAT3i in combination with the current standard of care methods provide an immunomodulatory response advantageous in the highly aggressive and recurring GBM disease model.

## Results

**Particle design, synthesis, and characterization.** SPNPs were prepared via electrohydrodynamic (EHD) jetting, a process that utilizes atomization of dilute solutions of polymers to produce well-defined NPs (Fig. 1a and Supplementary Fig. 1)[39–41]. Rapid acceleration of a viscoelastic jet in an electric field leads to a size reduction by several orders of magnitude facilitating rapid solvent evaporation and solidification of the non-volatile components into NPs. Here, the jetting solution comprised of HSA and a bifunctional OEG macromer (NHS-OEG-NHS, 2 kDa), which were mixed with therapeutic siRNA, polyethyleneimine (PEI, a siRNA complexing agent), and the tumor penetrating peptide, iRGD, prior to NP preparation. Similar to a step-growth polymerization, the OEG macromer was combined with albumin molecules through reaction with its lysine residues resulting in water-stable SPNPs. After EHD jetting and collection, the resulting SPNPs had an average diameter of 115 ± 23 nm in their dry state (Fig. 1b). Once fully hydrated, we observed that the average diameter of SPNPs increased to 220 ± 26 nm based on dynamic light scattering (DLS) measurements (Supplementary Fig. 2). The degree of NP swelling was controlled by varying the HSA-to-OEG ratios between 4:1 and 20:1 and the molecular weight of the OEG macromer between 1 and 20 kDa. An increase of the OEG content from 5 to 20% resulted in a reduction of SPNP swelling by 20%. The resulting SPNPs were stable for at least 10 days at 37 °C under physiological conditions; with no significant change in particle size or morphology (Supplementary Fig. 3). When exposed to mildly acidic conditions (pH 5.0), similar to those observed in endosomes of cancer cells, the diameters of SPNPs increased to 396 ± 31 nm (Fig. 1c). We note that defining particle properties, such as particle size, shape, and swelling behavior, was, within the margins of error, identical for fully loaded SPNPs, empty NPs and NPs loaded with siRNA and/or iRGD.

**In vitro cell uptake and siRNA activity.** Previously, co-delivery of the cell-penetrating peptide iRGD has increased tumor targeting for both, small drugs and iron oxide NPs[42]. The iRGD peptide has been shown to act in a three-stage process, first binding to αvβ3 and αvβ5 integrins, followed by a proteolytic cleavage step and a secondary binding event to neurophilin-1 (NRP-1), which activates an endocytotic/exocytotic transport pathway[43]. In the past, iRGD mediated tumor homing has been approached either in form of systemic co-delivery of iRGD with NPs[44,45] or by decorating the NPs with surface-bound iRGD[46,47]. In our case, the iRGD peptide is preloaded into SPNPs during EHD jetting to promote local release at the vicinity of the BBB. To investigate the intracellular fate of NPs and the effect of iRGD as a targeting ligand, in vitro uptake experiments were performed.

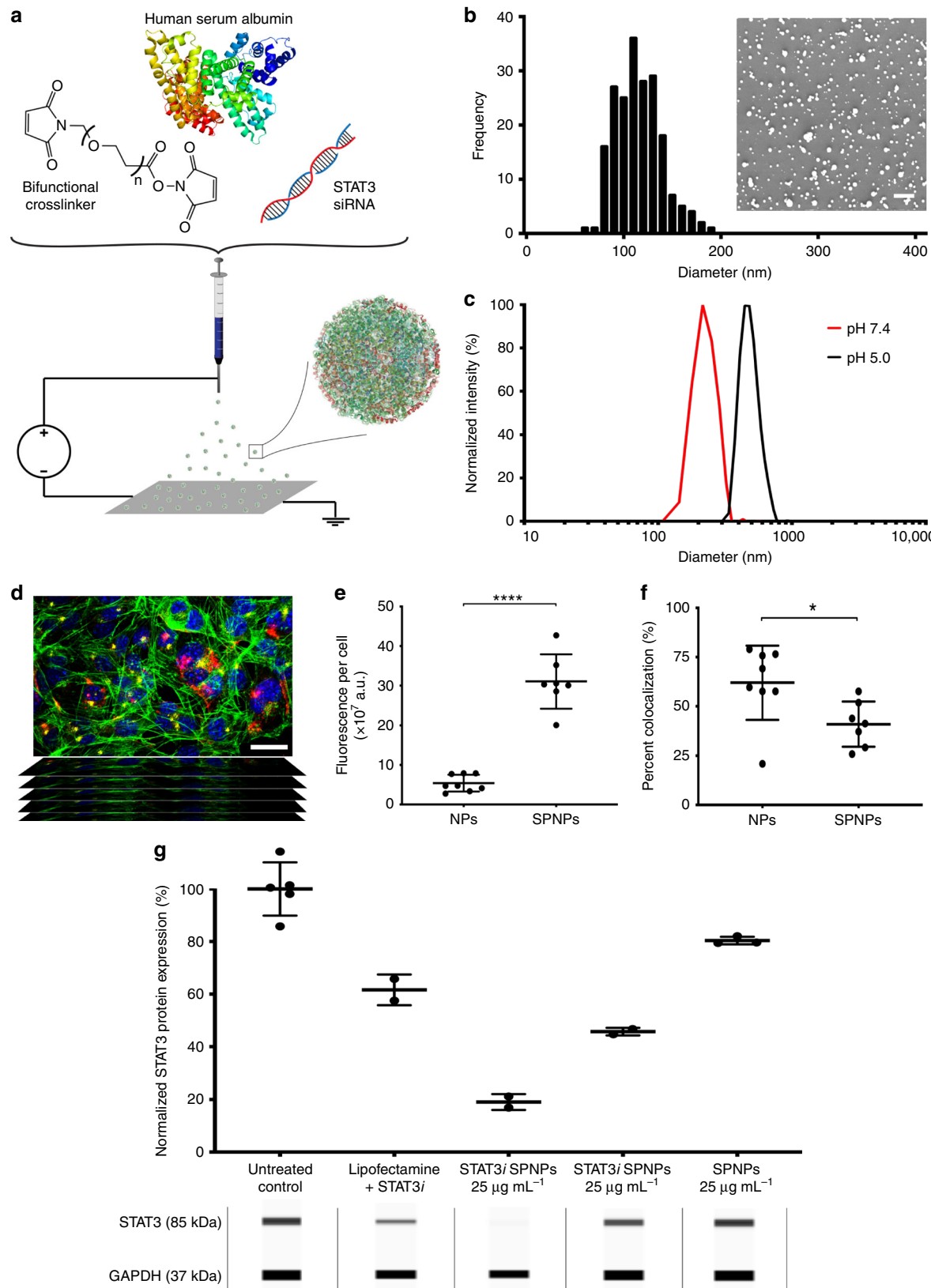

When iRGD-loaded SPNPs were incubated with GL26 glioma cells for a period of 30 min, particle uptake increased by ~5-fold compared to NPs without iRGD (Fig. 1e). Image analysis of 3D confocal images (Fig. 1d) using CellProfiler revealed that significantly less particles colocalized with lysosomes of GL26 glioma cells compared to NPs without iRGD (Fig. 1f). These data suggest that loading iRGD into SPNPs results in higher uptake and higher cytosolic SPNP concentrations in glioma cells. The pH-dependent swelling of SPNPs, along with the "proton sponge" effect previously postulated for PEI[48], may contribute to more effective particle escape from endocytotic vesicles, enhancing overall siRNA delivery to the cytosol and RNA-induced

**Fig. 1 STAT3 expression is effectively silenced in vitro by siRNA-loaded SPNPs. a** Schematic of the jetting formulation for crosslinked, STAT3i-loaded, iRGD-conjugated, targeted albumin NPs (STAT3iSPNPs). **b** Particle size characterization and analysis was performed using scanning electron microscopy (SEM). Average particle diameter, 115 ± 23.4 nm. Scale bar 1 μm. **c** Particles undergo swelling in their hydrated state and further swell at reduced pH. Average diameters: pH 7.4, 220 ± 26.1 nm. pH 5.0, 396 ± 31.2 nm. **d** Representative confocal z-stack image of cells cultured in the presence of SPNPs (blue = nucleus, green = actin, yellow = lysosomes, red = SPNPs). Composite images of cells incubated with SPNPs (with and without iRGD) were collected from a single independent experiment to study intracellular particle fate. Scale bar = 30 μm. **e, f** Quantification of particle uptake and lysosome-particle colocalization. **e** Local release of iRGD from SPNPs increases particle uptake in GL26 glioma cells by greater than five-fold (***$p < 0.0001$). **f** Internalized SPNPs colocalize with lysosomes to a lesser extent than untargeted particles (*$p = 0.0235$). Data are presented as mean values ± s.d. (SPNPs $n = 7$, NPs $n = 8$, independent composite z-stack images; two-tailed unpaired t-test). **g** STAT3 siRNA-loaded SPNPs significantly reduce in vitro expression of target protein in GL26 glioma cells compared to untreated and empty particle control groups. Data are presented as mean values ± s.d. (SPNPs, $n = 3$; Lipofectamine + STAT3i, STAT3i SPNPs (25 and 2.5 μg mL$^{-1}$, $n = 2$ biological replicates). Representative bands obtained with the ProteinSimple Wes instrument for both STAT3 (siRNA target protein) and GAPDH (loading control) are displayed for each experimental group.

silencing[49,50]. We note however, that the concentration of PEI in SPNPs is about 200-fold lower than what has previously been reported to be harmful to cells[51].

We next evaluated, if siRNA loaded into SPNPs during EHD jetting remains biologically active. First, siRNA loading and release from SPNPs was evaluated using a Cy3-labeled STAT3i surrogate. Utilizing stimulated emission depletion (STED) microscopy, we confirmed uniform distribution of siRNA throughout the entire NP volume (Supplementary Fig. 4). In vitro release of fluorescently tagged siRNA confirmed that 96% of the initial amount of siRNA was encapsulated into SPNPs; corresponding to a siRNA loading of 340 ng, or 25 pmol of siRNA per mg of SPNPs. Furthermore, we observed that ~60% of the encapsulated siRNA was released over the first 96 h, followed by a sustained release period progressing for 21 days (Supplementary Fig. 5). When albumin NPs were loaded with siRNA against GFP, SPNPs significantly suppressed GFP expression in mouse glioma cells transfected to express mCitrine (GL26-Cit, Supplementary Fig. 6) relative to control albumin NPs loaded with scrambled siRNA or free GFP siRNA that was delivered using lipofectamine as the transfection agent. Moreover, protein knockdown persisted significantly longer in the SPNP group than in lipofectamine-transfected cells (Supplementary Fig. 6). While the latter entered a recovery phase after two days and nearly returned to normal GFP levels by day five, cells treated with GFPi SPNPs showed sustained protein knockdown throughout the experiment. There were no significant differences in particle size, surface charge, or morphology between siRNA-loaded SPNPs and the control particles (Supplementary Fig. 7).

For SPNPs co-loaded with iRGD and STAT3i at concentrations of 2.5 and 25 μg mL$^{-1}$, we observed a significant reduction in total STAT3 protein expression relative to the untreated control group or empty SPNPs (Fig. 1g). Moreover, we observed a dose-dependent response in that a higher SPNP concentration resulted in ~2-fold further decrease in total STAT3 expression. No detectable signs of cytotoxicity were observed for any of the tested NP groups, which we attributed to the fact that the delivered siRNA concentrations were below the cytotoxicity limit observed for free STAT3 siRNA in GL26 cells (Supplementary Fig. 8). Based on these in vitro experiments, we chose an effective dose of 5 μg kg$^{-1}$ in subsequent animal studies.

In order to simultaneously evaluate particle stability and diffusion within the tumor microenvironment (TME), Alexa Fluor 647 SPNPs were administered intratumorally (Supplementary Fig. 9). Previous studies had demonstrated that iRGD enhances the penetration of co-delivered therapeutics throughout the tumor volume[42,43]. This becomes increasingly relevant in GBM where recurrence occurs locally in regions adjacent to the tumor resection cavity. Following intratumoral injection of SPNPs, NPs were widely distributed throughout the tumor volume extending from the injection site. Furthermore, we observed persistent fluorescent

signal associated with SPNPs within the tumor hours after intratumoral administration.

**Systemic delivery of SPNPs in an intracranial GBM model.** In the past, the BBB has been an unsurmountable delivery challenge for nanocarriers that are systemically administrated via IV injection[52,53]. To evaluate, if systemically delivered SPNPs can home to brain tumors, SPNPs loaded with Alexa Fluor 647-labeled albumin were prepared as described above. In the absence of large animal GBM models, we selected the very aggressive GL26 syngeneic mouse glioma model, which is known to exhibit histopathological characteristics encountered in human GBM[54], to evaluate GBM-targeting of SPNPs. In addition, this model features an uncompromised immune system, which was deemed to be essential, because of the prominent role that STAT3 plays in downregulation of the immune system. A dose of $2.0 \times 10^{13}$ SPNPs was delivered to GBM-bearing mice via a single tail vein injection seven days after glioma cell implantation (GL26-Citrine, approximate tumor size: 10 mm$^3$, Supplementary Fig. 10) in the right striatum of the mice (Fig. 2a). After 4 or 24 h, animals were perfused, and brains were collected, sectioned, and stained with F4/80 (a marker for macrophages), prior to confocal imaging. We also observed that SPNPs were taken up by other organs, such as liver, kidney, spleen, and the lungs (Fig. 2d). A significant number of SPNPs appeared to have crossed the BBB and were identified within the brain TME at both time points (Fig. 2b). Tumor targeting was markedly increased after 24 h, hinting towards the possibility that secondary transport processes, such as transcytotic or immune-cell-mediated BBB pathways, are contributing to the brain-homing of SPNPs, in addition to the direct targeting of the brain endothelium by circulating NPs. The notion of a multi-variant transport mechanism is consistent with our finding that SPNPs were localized inside of tumor cells (green) and macrophages (red), suggesting that both cell types can internalize SPNPs (Fig. 2b).

Using the same intracranial tumor model, GBM specific biodistribution of SPNPs was assessed. Tumor-bearing mice were injected three times (7, 10, and 13 days post implantation, DPI) with Alexa Fluor 647-labeled SPNPs or NPs without iRGD (Fig. 2c). In addition, normal mice (non-tumor bearing) were subjected to the same regimen to delineate tumor-specific characteristics. After 14 days, normal, non-tumor bearing mice and tumor-bearing mice were euthanized, their main organs were collected, and the NP distribution was analyzed via fluorescence imaging (Fig. 2d). In both GBM-bearing and non-tumor bearing mice, significantly more SPNPs were observed in the brain compared to iRGD-free NPs. As expected, SPNPs also accumulated in the lungs and liver—independent of the particular experimental group. The former can be attributed to being the first capillary bed the NPs would encounter following intravenous

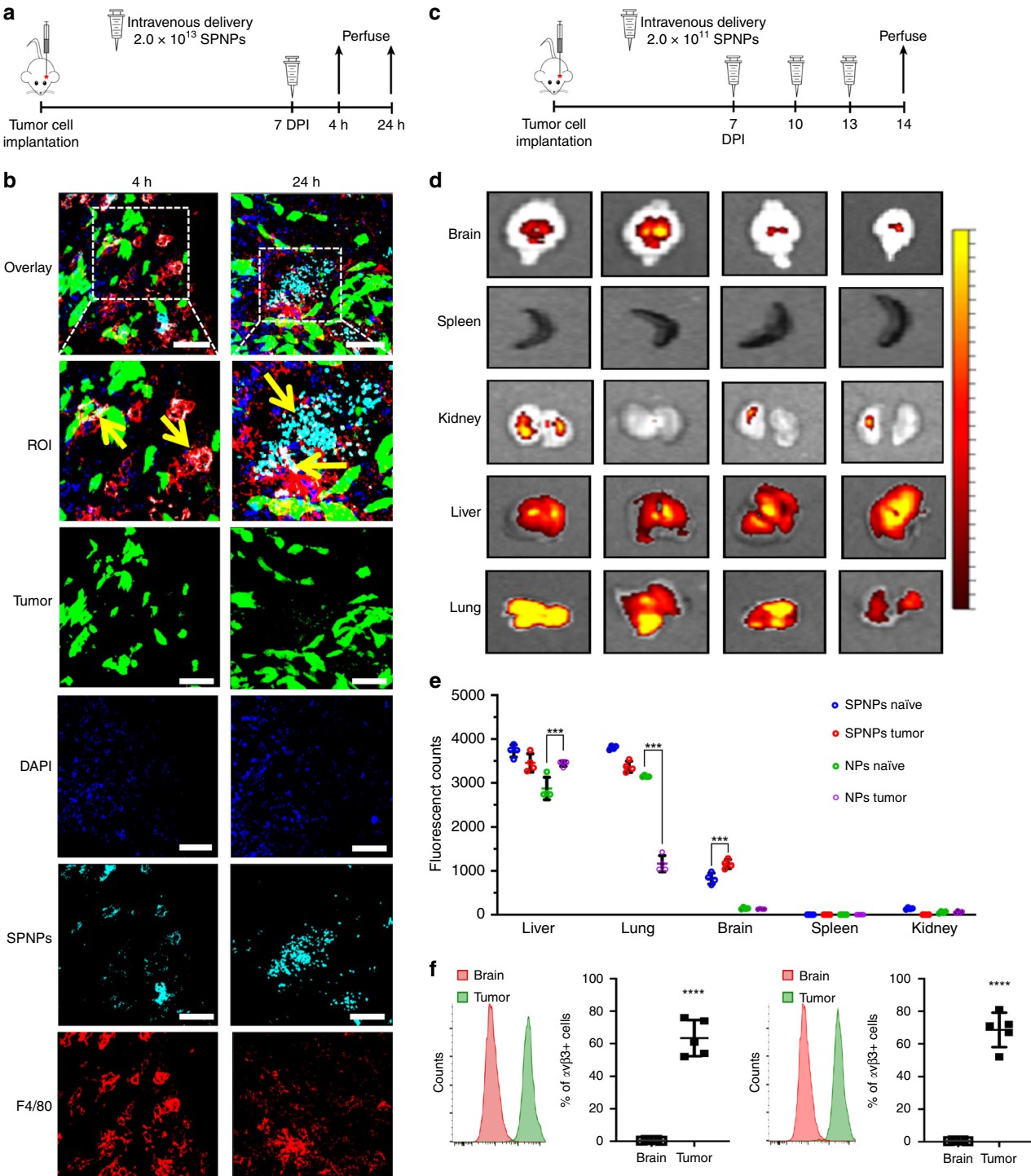

**Fig. 2 in vivo brain targeting and biodistribution of SPNPs. a** Timeline for the tumor-targeting study. Mice were IV administrated a single dose of 2.0 × 10^13 SPNPs or empty NPs (no iRGD) via the tail vein seven days post GL26 tumor cells implantation. Confocal imaging of sectioned brains was performed 4 and 24 h post particle administration. **b** Alexa Fluor 647 labeled SPNPs (cyan) colocalize (indicated with yellow arrows) with macrophages (red) and tumor cells (green, mCitrine). Notably less NPs are observed in the tumor microenvironment 4 h post systemic delivery compared to 24 h. Representative images from a single experiment consisting of three biological replicates per group are displayed. Scale bars = 50 μm. **c** Timeline representation of the biodistribution study. Mice were IV administered 2.0 × 10^11 SPNPs or empty NPs 7, 10, and 13 days post tumor cell implantation or saline injection. **d** Fluorescence imaging of tumor-naive and tumor-bearing mice organs sacrificed at 24 h post final NP delivery. **e** Quantitative analysis of NP biodistribution within the tumor and peripheral organs. Data are presented as mean values ± s.d. (n = 4 biological replicates, two-way ANOVA; ***p < 0.0001). **f** Quantitative flow cytometry results of αvβ3 and αvβ5 integrin expression in normal brain tissue and GL26 tumors. Data are presented as mean values ± s.d. (n = 5 biological replicates; two-tailed unpaired t-test; ****p < 0.0001).

injection, while the latter represents the primary route of clearance for NPs measuring 10–250 nm in diameter[55]. Brain accumulation of SPNPs loaded with iRGD was higher for both naïve and GBM-bearing mice compared to iRGD-free NP groups. When directly comparing SPNP localization within the brain compartment, the accumulation of iRGD-loaded SPNPs was 40% higher in tumor-bearing brains (Fig. 2e) than in non-tumor bearing mice.

Next, we evaluated whether the increased accumulation of SPNPs in the brain of tumor-bearing mice is mediated by integrin expression on tumor cells. Specifically, we focused on the relative expression of αvβ3 and αvβ5 integrins, because these ligands have been shown to play pivotal roles in the iRGD-induced accumulation of NPs and small drugs in tumors[31], and are overexpressed in gliomas[32]. Using the GBM model and dosing schedule from the biodistribution studies, brains from GBM tumor-bearing mice were collected at 23 DPI. Normal brain and tumor tissue were dissected from the brain, processed, and stained with αvβ3 and αvβ5 antibodies for flow cytometry analysis. More than 60% of the GBM tumor population expressed αvβ3 and αvβ5 integrins at high levels, while normal brain cells showed minimal expression of these proteins. These results, along with the observed differences in brain accumulation in the biodistribution study, appear to be consistent with the previously postulated hypothesis that iRGD promotes the accumulation of SPNPs in the brain[43].

**In vivo survival studies.** Previous data indicate that the Signal and Transducer of Activation 3 (STAT3) transcription factor is a hub for multiple signaling pathways which mediate tumor progression and immune functions[6,56–58]. There are no effective delivery strategies of anti-STAT3 therapeutics to the brain[6]. We observed that systemic delivery of a single dose of STAT3i SPNPs to GMB bearing mice significantly increased their MS (Supplementary Fig. 11). To further test the efficacy of SPNPs in vivo, GBM-bearing mice were treated intravenously with multiple doses of STAT3i SPNPs over the course of a three-week treatment regimen (Supplementary Fig. 12). After tumor implantation, the MS of untreated mice was about 28 days. In mice that received multiple doses of empty SPNPs, the MS remained unaltered (28 days). In contrast, when SPNPs loaded with STAT3i were administered, the MS increased to 41 days, a statistically significant increase of 45%. Delivery of the same doses of free STAT3i resulted in a modest extension of MS by 5 days, which is likely too low to elicit a significant therapeutic effect. The low efficacy of free STAT3i can be explained by the rapid degradation of genetic material following systemic administration—in addition to siRNA's inability to cross the BBB[59].

Encouraged by the prospect of a NP formulation for STAT3i delivery with significant in vivo efficacy, we combined STAT3i SPNPs with the current standard of care, i.e., focused radiotherapy (IR). Previous studies have identified a direct correlation between STAT3 overexpression and radioresistance in other cancers[60,61], suggesting that its knockdown could contribute to enhanced efficacy. We thus established a treatment protocol that combined the previously evaluated multi-dose regimen with a repetitive, two-week focused radiotherapy regimen (Fig. 3a)[62,63]. Once GBM tumors had formed, mice received seven doses of STAT3i SPNPs over the course of a three-week period. During each of the first two weeks of therapy, mice also received five daily 2 Gy doses of IR for a total of ten treatments (Fig. 3a). Experimental groups included mice that received either STAT3i SPNPs, empty SPNPs, free STAT3i, or saline with or without combined radiotherapy. In all cases, the addition of radiotherapy increased the MS, with IR alone resulting in a MS extension from

28 to 44 DPI (Fig. 3b). Combining IR with empty SPNPs did not further alter the MS. Consistent with our previous experiment, free siRNA provided a slight, statistically significant benefit, where the MS was increased to 58 DPI, when combined with IR (Fig. 3b, brown line). However, the most significant effect was observed for the combination of STAT3i SPNPs with IR. Of the eight mice in this group, seven reached the standard long-term survivor time point of 90 DPI and appeared to be completely tumor-free thereafter (Fig. 3b, blue line). The single mouse receiving this treatment that did not reach long-term survival was moribund at 67 DPI, living longer than any other non-surviving subject from all other groups.

In order to characterize the effects of the combined treatments, additional studies were performed. Following the same treatment outlined in Fig. 3a, we elucidated the expression of STAT3 and its active phosphorylated form, pSTAT3, in the brain tissues (Fig. 3c). As expected, the greatest reduction in both the total and phosphorylated form of the protein was found in the STAT3i SPNP group. Greater than 50% reduction in total STAT3 protein was present in GBM bearing mice treated with STAT3i SPNP, compared to the saline-treated control. Greater than 10-fold reduction in pSTAT3 levels were observed in GBM bearing mice treated with STAT3i SPNP, compared to the saline-treated control. In contrast, the total STAT3 levels were relatively unchanged in both, the free STAT3i and empty SPNP groups, compared to saline control. Here, pSTAT3 was increased by 110% in the cohort receiving empty SPNPs, suggesting a shift in the balance of the two protein forms, perhaps due to a localized upregulation of kinase activity.

Immunohistochemistry (IHC) analysis was used to compare the brains of long-term survivors to other treatment groups. Figure 3d shows a direct comparison of the brain of a long-term survivor and that of a control animal that did not receive treatment. Hematoxylin and Eosin (H&E) staining clearly shows the presence of a fully developed tumor localized within a single hemisphere of the saline-treated mice. Conversely, mice treated with STAT3i SPNP + IR showed no evidence of intracranial tumor (Fig. 3d, top). Moreover, no apparent regions of necrosis, palisades or hemorrhages were present in these animals 90 DPI after receiving a full course of therapy. Myelin basic protein (MBP) staining was performed to assess the integrity of myelin sheaths, an indicator for the disruption of surrounding brain architecture. No apparent changes in myelin sheath morphology were observed in mice that received the combined STAT3i SPNP + IR treatment when compared to the cancer-free right faces of mice in the saline-treated control group (Fig. 3d, middle). In addition, CD8 T cells were sparse in the TME (Fig. 3d, bottom) and their total number was significantly reduced in STAT3i SPNP + IR treated mice compared to the saline-treated control group (Supplementary Fig. 13)—indicating a lack of inflammation response due to the treatment.

To further investigate the potential role of the adaptive immune system, we more closely examined the population of CD8 T cells within the TME via flow cytometry. We established tumors in mice using GBM cells that harbored a surrogate tumor antigen ovalbumin (OVA) and compared the responses elicited by the various treatment formulations (Fig. 3e). The OVA-based GBM model was utilized to assess the frequency of tumor antigen-specific T cells within the GBM microenvironment, due to the availability of OVA-specific MHC tetramers. Tumor-specific T cells were characterized by staining for the SIINFEKL-H2K$^b$-OVA tetramer, an OVA cognate antigen within the CD8 T cell population. Tumor-specific CD8 T cells (CD3+/CD8+/SIINFEKL-H2K$^b$ tetramer) within the STAT3i SPNP + IR group were increased by two-fold compared to all other groups (Fig. 3f, top). Staining the same population of cells with interferon-γ

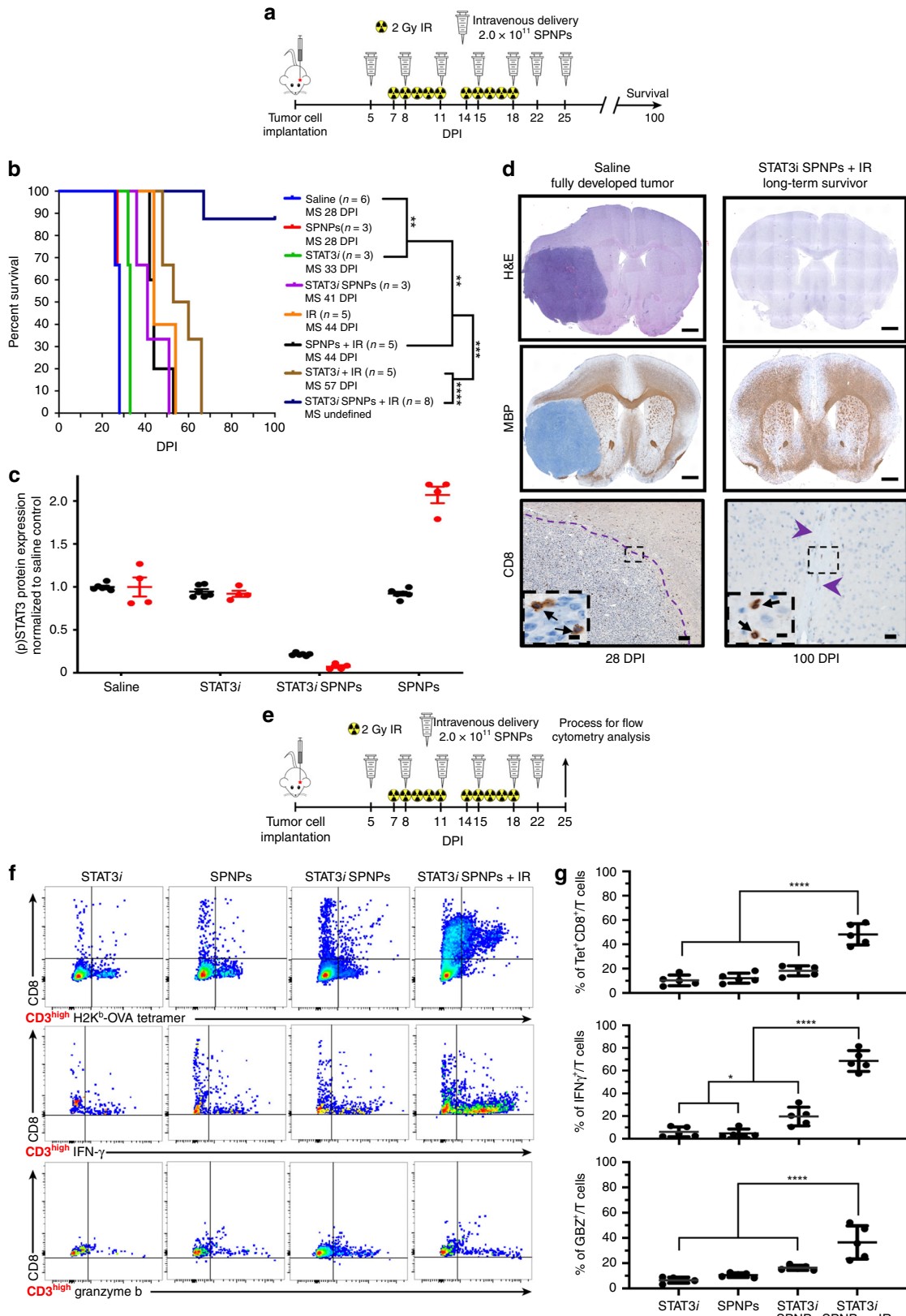

(IFN-γ) and granzyme B (GZB) revealed a two-fold increase in cytotoxic T cells in the TME (Fig. 3f, middle and bottom) in the STAT3i SPNP + IR group relative to all other groups. Flow cytometry gating strategies used within this study are presented in Supplementary Fig. 14. Taken together with the increased MS of GL26 GBM mice in the survival study, these results suggest a

robust anti-GBM response elicited by the combined STAT3i SPNP + IR therapy that is likely contributing to the observed therapeutic success.

Recognizing that significant SPNPs accumulated in the liver (Fig. 2d), complete blood cell counts, serum biochemistry, and liver histopathological analysis were performed to examine

**Fig. 3 STAT3i SPNPs + IR results in increased survival and primes an adaptive immune response. a** Timeline of treatment for the combined NP + IR survival study. **b** Kaplan–Meier survival curve. Significant increase in median survival is observed in all groups receiving IR. Mice (7/8) treated with STAT3i SPNPs + IR reach long-term survival timepoint (100 DPI) with no signs of residual tumor (Log-rank (Mantel-Cox) test; ****$p < 0.0001$, ***$p < 0.001$, **$p < 0.01$). **c** Quantified STAT3 expression for resected brains from the survival study, brains were collected when mice displayed signs of neurological deficits. A significant reduction is STAT3 (black) and pSTAT3 (red) expression was observed in the STAT3i SPNP cohort relative to untreated control. Both soluble STAT3i and empty SPNPs (with no siRNA) did not have high total STAT3 expression but they had increased levels of active phosphorylated STAT3 expression (pSTAT3). Data are presented as mean values ± s.d. relative to untreated control ($n = 2$ biological replicates for each group). Due to minimal biological samples available, three technical replicates were performed on each sample to validate the experimental protocol and rule out measurement error. **d** IHC staining for untreated control and STAT3i SPNP + IR long-term survivor. (Top) H&E staining shows the fully formed tumor in the saline control group (28 DPI). When treated with the combination of STAT3i SPNPs + IR, no tumor or signs of necrosis were observed. (Middle) MBP staining shows preserved brain structures with no apparent changes in oligodendrocyte integrity in mice that received STAT3i SPNPs + IR treatment compared to the saline control. Scale bars = 1 mm. (Bottom) CD8 staining shows no overt inflammation in mice that received STAT3i SPNPs + IR treatment compared to the saline control. Representative images from a single experiment consisting of three biological replicates per group are displayed. Scale bars = 100 μm (inset, 20 μm). **e** Timeline of TME immune population by flow cytometry. **f** Flow cytometry analysis of CD8 cells in the TME. Representative flow plots for each group are displayed. **g** Quantitative analysis of tumor-specific CD8+ T cells within the TME. GL26-OVA tumors were analyzed by staining for the SIINFEKL-K$^b$ tetramer. Activation status of CD8+ T cells within the TME was analyzed by staining for granzyme B (Gzb) and IFNγ after stimulation with the tumor lysate. Data are presented as mean values ± s.d. ($n = 5$ biological replicates; one-way ANOVA and Tukey's multiple comparison tests; ****$p < 0.0001$).

potential off-target side effects of the combined therapeutic strategy. Systemic toxicity of STAT3i SPNPs + IR treatment was evaluated following the treatment schedule indicated in Fig. 4a. No significant differences in the cellular components of the blood were noted in complete blood cell counts analysis for animals receiving SPNP, STAT3i, STAT3i SPNP, or STAT3i SPNP + IR treatment compared with animals in the saline treatment group (Fig. 4b–i). Moreover, there was no significant difference in important biomarkers involved in the kidney (creatinine, blood urea nitrogen) and liver (aminotransferase, aspartate aminotransferase) physiology for animals receiving SPNP, STAT3i, STAT3i SPNP, or STAT3i SPNP + IR treatment compared with animals in the saline treatment group (Supplementary Table 1), indicating that no overt adverse side-effects occurred in these tissues.

In addition, independently conducted pathological analysis of potential side effects affecting the livers of mice treated with STAT3i SPNP + IR therapy revealed minimal to mild mononuclear pericholangitis across all groups and it was characterized as spontaneous background rather than a direct result of the applied therapy. (Fig. 4j). In all treatment groups, with the exception of the saline-treated control, minimal to mild coagulative necrosis was present. In the treatment group that received free STAT3i, one animal displayed multiple foci of coagulative necrosis, which distinguished it from all other animals in the entire study cohort, including those from the combined STAT3i SPNP + IR group, where the regions of necrosis were generally small and were deemed not to induce biologically significant adverse effects on liver function.

Next, SPNP-induced immune responses were assessed using a modified enzyme-linked immunosorbent assay (ELISA, Supplementary Fig. 15). To avoid a species-to-species mismatch due to the use of HSA in mice, otherwise identical NPs were synthesized, in which HSA was replaced with mouse serum albumin (MSA). No circulating antibodies specific to MSA SPNPs were observed in any of the treatment groups indicating neglectable immunogenicity against any of the individual components of SPNPs, such as OEG, STAT3i, iRGD, or PEI (Fig. 4k). As expected, replacing MSA with HSA resulted in elevated levels of HSA antibodies for both, STAT3i SPNP and empty SPNP treatment groups (Fig. 4l). Free STAT3i therapy did not induce this same response suggesting that antibodies were formed in response to the exposure of the NPs rather than the active therapeutic ingredient.

Taken together, our results indicate that sequential intravenous administration of STAT3i SPNP in combination with radiation

does not cause systemic toxicity. It is well documented that the mononuclear phagocyte system, which includes the liver and kidney tissue, takes up intravenously injected NPs[64]. Levels of biochemical parameters including creatinine, blood urea nitrogen, aminotransferase, and aspartate aminotransferase in the serum are good indicators of acute inflammation in liver and kidneys. In our study, there were no inflammatory reactions caused by STAT3i SPNP + IR therapy (Fig. 4 and Supplementary Table 1), indicating that our treatment strategy does not cause short-term toxicity. Further studies will need to be performed to illuminate potential long-term side effects of the STAT3i SPNP + IR therapy.

Flow cytometry analysis of tumor-infiltrating macrophages and conventional dendritic cells (cDCs:CD45+/CD11c+/B220−) was used to compare treatment groups containing free STAT3i, empty SPNP, and STAT3i SPNPs in combination with IR (Figs. 5a and b). Co-staining of CD45+ cells with F4/80 and CD206 antibodies was used to establish a subpopulation of tumor-associated macrophages (TAMs). Within the TAM population, both, M1 (CD45+/F4/80+/CD206−) and M2 (CD45+/F4/80+/CD206+) macrophage phenotypes, were identified for all cohorts, but their relative abundance was significantly different in the STAT3i SPNP + IR group compared to all other groups. In the STAT3i SPNP + IR group, M1 macrophages were increased by 2.5-fold (Fig. 5c, left), whereas the number of M2 macrophages was decreased by three- to four-fold (Fig. 5c, middle). These findings are consistent with the notion that the STAT3i SPNP + IR treatment selectively decreases the immune suppressive M2 macrophage subpopulation. In addition, antigen presentation by cDCs was significantly higher in animals receiving STAT3i SPNPs compared to free siRNA and empty SPNPs (Fig. 5c, right). IR treatment further elevated this effect resulting in the largest cDC population for the STAT3i SPNP + IR group. Co-staining of CD45 and F4/80$^{high}$ cells with CD206 and Arg1 antibodies in the STAT3i SPNP + IR group confirmed that the vast majority of TAMs were of the M2 phenotype (Supplementary Fig. 16). Among all TME CD45+ immune cells, only M2 macrophages displayed the far-red Alexa Fluor 647 signal indicative of SPNPs suggesting that immune suppressive M2 macrophages are the primary TME-based immune cells that internalize SPNPs (Supplementary Fig. 17).

Finally, we analyzed the maturation status of DCs in the draining lymph nodes (dLNs) of free STAT3i, SPNPs, STAT3i SPNPs and STAT3i SPNPs + IR treated GL26-OVA tumor-bearing mice using flow cytometry. To examine the effect of

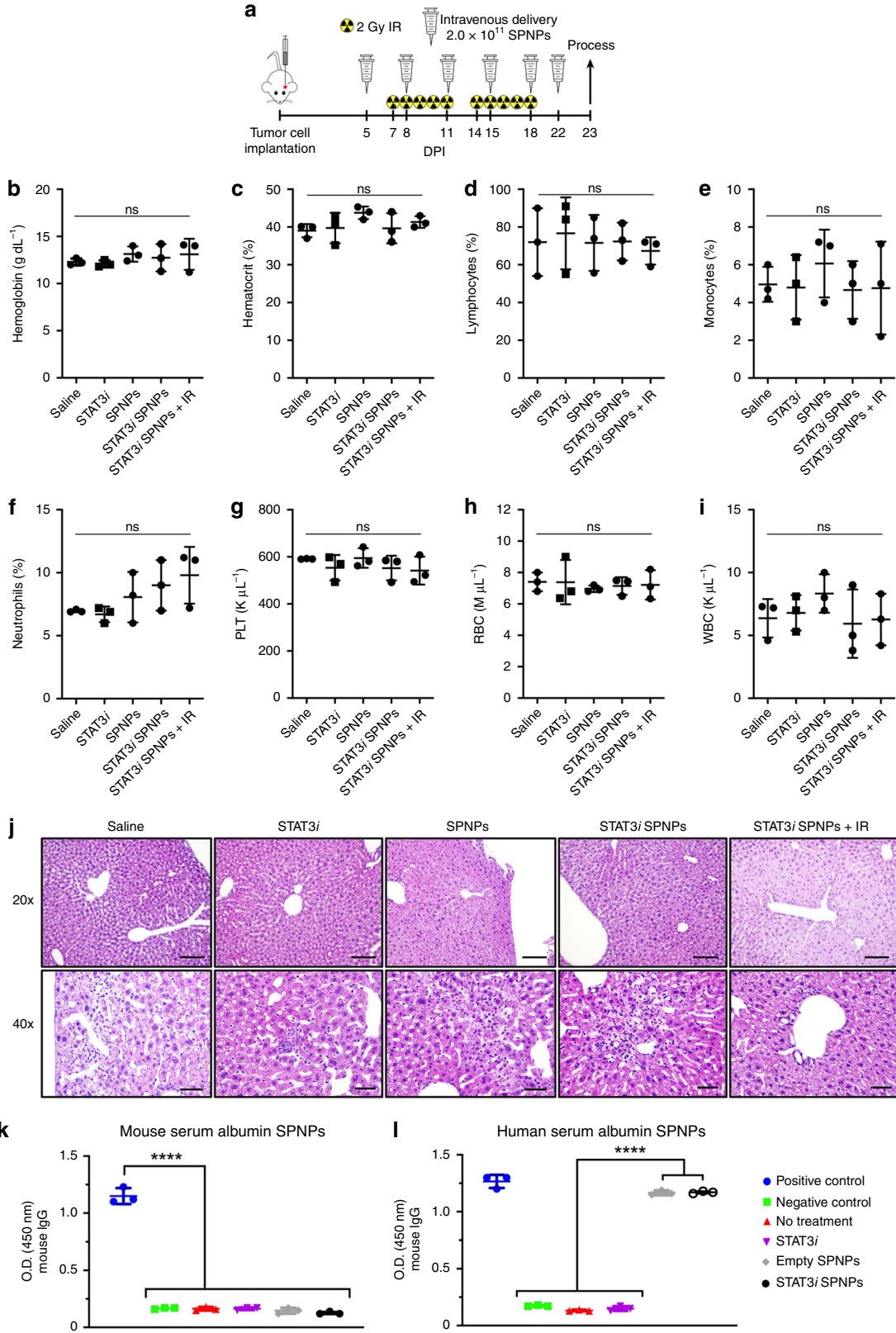

SPNPs on DC activation in the dLNs, the expression of MHC II was assessed. We observed an increase in the frequency of DCs: CD45+/ CD11c+/ MHC II + (~1.5-fold, $p < 0.0001$), in the dLN of STAT3$i$ SPNPs treated mice compared to free-STAT3$i$ and SPNP treated mice. This was further enhanced in the presence of IR by ~1.3-fold ($p < 0.0001$) (Supplementary Fig. 18). These data

suggest that STAT3$i$ SPNPs in combination with radiation induces the activation of DCs by enhancing the expression of MHC II, which is involved in antigen presentation.

**Tumor rechallenge study**. The current standard of care approaches, including surgical resection combined with focused

**Fig. 4 STAT3i SPNPs + IR results in no abnormal toxicity. a** Timeline of treatment to assess the effect of STAT3i SPNP + IR treatment. Blood and liver samples were collected from GL26 GBM bearing mice treated with saline, STAT3i, SPNP, STAT3i SPNP or STAT3i SPNP + IR at 23 DPI following complete therapeutic regimen. **b–i** For each treatment group, levels of (**b**) hemoglobin, (**c**) hematocrit, (**d**) lymphocytes, (**e**) monocytes, (**f**) neutrophils, (**g**) platelet, (**h**) red blood cell (RBC), and (**i**) white blood cell (WBC) counts were quantified. Data are presented as mean values ± s.d. ($n = 3$ biological replicates; one-way ANOVA and Tukey's multiple comparison tests; ns = $p > 0.05$). **j** Histology performed on resected livers following a complete treatment of GMB tumor-bearing mice find isolated regions of mild coagulative necrosis deemed to be well-contained and therefore would not induce a biological effect on liver function. In all groups, with the exception of the saline-treated control, signs of hepatocellular necrosis were observed. This is attributed to water or glycogen accumulation in hepatocytes associated with a change in energy balance rather than a degenerative change. Representative images from a single experiment consisting of independent biological replicates are displayed. Scale bars = 100 μm (20×), 50 μm (40×). **k, l** Following complete STAT3i SPNPs + IR treatment, circulating antibodies against (**k**) mouse serum albumin nanoparticles were not observed in serum of GL26 tumor-bearing mice. When exchanging (**l**) human serum albumin in the formulation, maintaining all other components, measurable serum HSA-specific antibody levels are observed. Data are presented as mean values ± s.d. ($n = 3$ biological replicates; one-way ANOVA and Tukey's multiple comparison tests; ****$p < 0.0001$).

radiation and the chemotherapeutic temozolomide, have been used to treat primary GBM tumors. However, owing to the aggressive and infiltrative nature of GBM, these patients, as a rule, experience recurrence contributing to the high mortality and dismal survival rates. Based on the encouraging immune response observed in our survival study, we chose to rechallenge survivors from the STAT3i SPNP + IR treatment group. Tumors were implanted in the contralateral hemisphere of mice that were previously cured by the STAT3i SPNP + IR therapy. These mice did not receive any additional intervening therapy (Fig. 5d). As a control, normal mice were also implanted with tumors at the same timepoint and likewise received no treatment. As expected, the control group saw rapid tumor growth, increased signs of disease and had a MS of 27 DPI. Despite not receiving any additional treatment, all rechallenged mice survived to a second long-term survival point of 90 DPI (relative to the second tumor implantation, 180 days post initial tumor implantation) (Fig. 5e). IHC analysis of the brains yielded comparable results (Fig. 5f). H&E staining clearly showed the formation of a fully developed tumor mass in the control group, while members of the rechallenged cohort displayed no regions of necrosis, palisades or hemorrhages in either hemisphere (Fig. 5f, top). MBP staining confirmed that there was no overt disruption of the surrounding brain architecture (Fig. 5f, middle). Lastly, the presence of CD8 T cells was observed to be fivefold lower (Supplementary Figure 19) in the STAT3i SPNP rechallenge group compared to the control (Fig. 5f, bottom). Importantly, we found no adverse effects in the brains of rechallenged survivors. Previous work has demonstrated that glioma cell death is associated with the concomitant release of antigens and damage-associated molecular patterns (DAMPs) leading to tumor antigen-specific T cell expansion and adaptive anti-glioma immunity[65]. Our findings suggest a similar involvement of an adaptive immune response that appears to guard against secondary tumors; an essential condition of any successful GBM therapy that will require long-term eradication of migrating and resistant CSCs, typically missed by traditional therapies.

## Discussion
Despite research efforts made over the past several decades, GBM remains one of the most aggressive forms of cancer with characteristically high levels of recurrence and low MS rates[2]. While the identification of key biological pathways has yielded promising approaches towards effective therapeutic targets, for the most part, these have ultimately resulted in marginal advances. In the case of STAT3, which is involved in multiple signaling pathways related to GBM tumor progression and immune response[7], previous studies have demonstrated positive therapeutic effects in vitro and in peripheral tumors, but small molecule inhibitors of STAT3 proved to be ineffective in intracranial models of the disease[6]. This can be directly attributed to the inability of therapeutics to penetrate the

BBB and reach the tumor in clinically relevant concentrations. Results from our study show a similar trend: while we observe effective in vitro knockdown of STAT3 using free siRNA, this does not translate into reduced protein expression in an intracranial GBM model. As a result, the minimal therapeutic effect is observed in mice treated with STAT3i, alone or in combination with ionizing radiation (Fig. 3).

A wide range of experimental approaches have been proposed in recent years to address challenges associated with drug delivery to the brain and are based on either systemic or intracranial delivery strategies. To date, experimental studies that report even modest levels of efficacy are scarce and almost exclusively require invasive convection-enhanced intracranial delivery[18,25]. As for systemic delivery, the situation is even more troublesome with only a couple of studies reporting long-term survivors[12,29].

Contrary to previous approaches utilizing carriers made from synthetic materials, we took our inspiration from nature and the ability of native proteins and viral particulates to target and get transported across the BBB. HSA, when polymerized, forms water-stable NP. Previous studies have shown albumin-based nanocarriers to engage cell-surface receptors overexpressed in glioma cells and tumor vessel endothelium, SPARC, and gp60[34,35]. We observe that the resulting particles undergo pH responsive swelling, suggesting a flexible and dynamic architecture contrary to more rigid synthetic polymer-based NP. In addition, the amphiphilic protein structure makes it biochemically compatible with small molecule drugs, biological therapeutics (siRNA), and targeting peptides.

The cell-penetrating peptide iRGD has been shown to increase the accumulation of both small molecule drugs and NP drug delivery systems when co-delivered or conjugated to particle surfaces[42]. Shown to initially bind to αvβ3 and αvβ5 integrins, iRGD initiates an endocytotic/exocytotic transport mechanism responsible for its tissue penetrative properties[43]. Here, we observe increased levels of both αvβ3 and αvβ5 integrins in GBM tumor-bearing mice and a corresponding increase in tumor accumulation of SPNPs (Fig. 2).

Encouraged by results showing the accumulation of SPNPs in GBM tumors we sought to evaluate the therapeutic efficacy of STAT3i SPNPs in combination with focused radiotherapy. In the highly aggressive GBM GL26 model, a significant increase in MS is observed in mice treated with the combined therapy with 87.5% of mice reaching the long-term survival timepoint. In these mice, we observed significantly reduced levels of STAT3, no apparent residual tumors, normal brain architecture, and a lack of inflammation in response to the treatment. We observed increases in both tumor-antigen specific CD8 T cells in the brain TME along with a decrease in immune suppressive M2 macrophages suggesting the activation of an anti-GBM immune response (Fig. 4). Finally, we observed minimal signs of toxicity in the liver and no significant differences in the cellular components

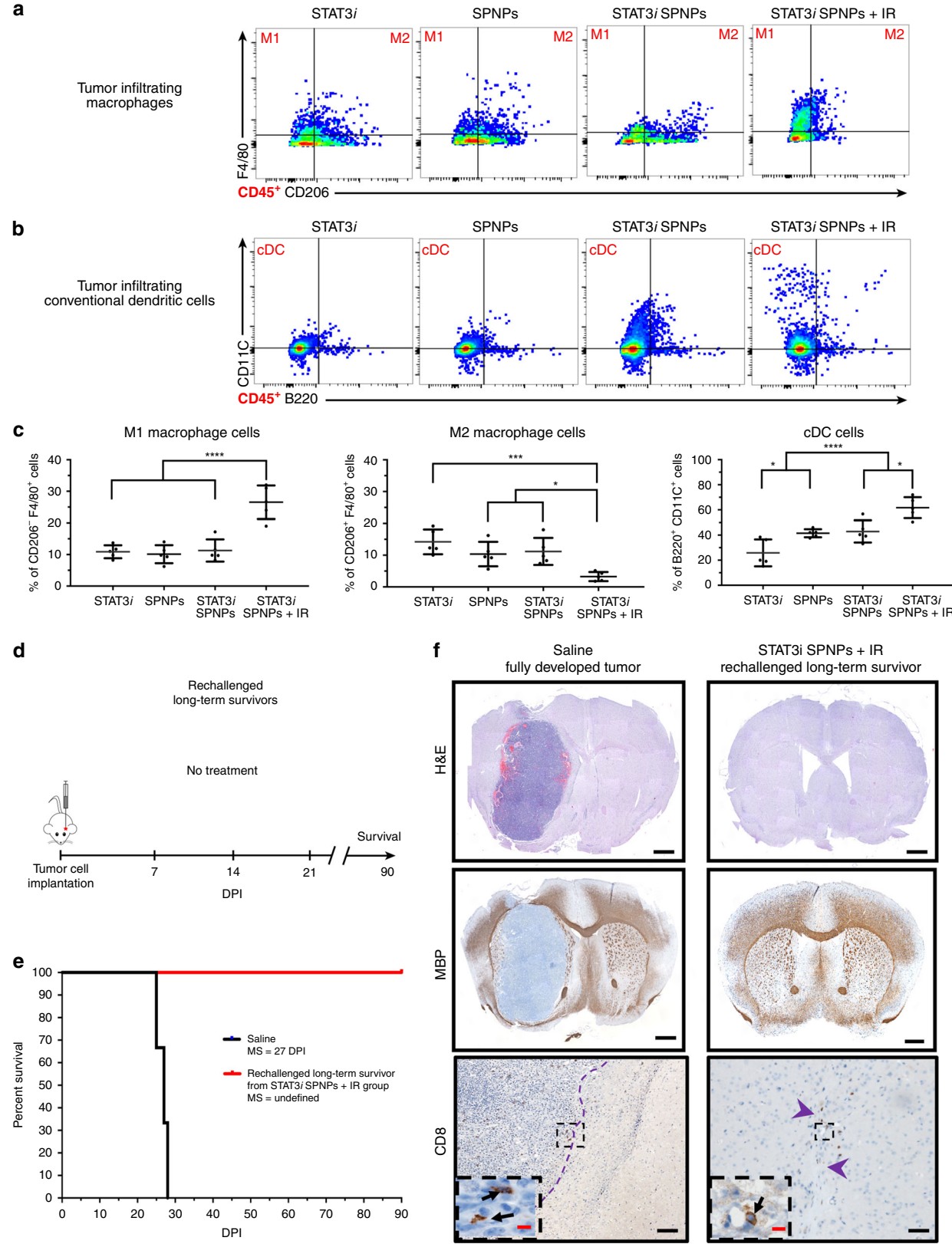

of blood relating to kidney and liver function suggesting no overt off-target side effects occurred as a result of the treatment.

Patients treated with conventional therapies including chemotherapeutics, radiation, and surgical resection, commonly experience recurrence in surrounding tissues contributing to GBM's high mortality rate. To further explore the observed

immune response, mice reaching the long-term survival timepoint were rechallenged with a second tumor in the contralateral hemisphere. Incredibly, in the absence of therapeutic intervention, all rechallenged mice survived to a second long-term survival timepoint. Rechallenged mice showed no overt signs of residual tumor, regions of necrosis, or disruption of the

**Fig. 5 SPNPs protect against GBM rechallenge. a, b** Flow analysis of tumor-infiltrating (**a**) macrophage and (**b**) conventional dendritic cell (cDCs) populations in the TME following NP + IR treatment regimen. Representative flow plots for each group are displayed. **c** Quantitative analysis of the immune cellular infiltrates showed a shift in the relative macrophage (M1vs. M2) present in the TME. In the free siRNA, empty SPNPs, and STAT3*i* SPNPs groups no significant change in the macrophage population were observed. Conversely, STAT3*i* SPNPs + IR treatment induces both a surge of ~2.5-fold increase in M1 population and a sharp 3- to 4-fold decrease in M2 macrophages. Among cDCs, progressively larger numbers of the cell population were observed moving from free siRNA to SPNPs groups. The combined treatment of STAT3*i* SPNPs with IR displayed the highest number of cDCs in the brain TME. Data are presented as mean values ± s.d. (= 5 biological replicates; one-way ANOVA and Tukey's multiple comparison tests; ****$p < 0.0001$, ***$p < 0.001$, *$p$ (M2) = 0.028, 0.013, *$p$(cDC) = 0.038, 0.011). **d** Timeline for rechallenging the long-term survivor for STAT3*i* SPNPs + IR survival study rechallenged, where following tumor implantation, no further treatment was provided. **e** Kaplan–Meier survival curve shows all rechallenged survivors reach a second long-term survival timepoint of 90 DPI in the absence of any therapeutic interventions. **f** H&E (Top, Scale bars = 1 mm), MBP (Middle, Scale bars = 1 mm), and CD8 (Bottom, Scale bars, 100 μm (inset 20 μm)) IHC staining comparing the brains of untreated and rechallenged long-term survivors. Representative images from a single experiment consisting of three biological replicates per group are displayed. No overt signs of remaining tumor, necrosis, inflammation, or disruption of normal brain architecture was observed in rechallenged long-term survivors from STAT3*i* SPNPs treatment group.

surrounding brain architecture (Fig. 5). Together, these studies further suggest the activation of an adaptive immune response, potentially capable of eradicating secondary tumors resulting from the aggressive and infiltrative nature of GBM.

SPNPs combine the biological benefits of proteins with the precise engineering control of synthetic NPs to yield (i) high efficacy (87.5% long-term survivors in a very aggressive intracranial tumor model), (ii) effective tumor delivery using systemically administered NPs, and (iii) possibilities towards long-term eradication of resistant cancer cells using immunomodulatory protein NPs. While protein-based NPs have been a fairly uncharted area of research, this study demonstrates that synthetic NPs that use proteins as structural building blocks may pave a viable route towards clinical cancer therapy implementation. We studied the NP-mediated delivery of a siRNA against STAT3 (STAT3*i*), but the SPNP platform could be adopted, after further development and preclinical testing, for delivery of small-molecule drugs, other siRNA therapies, or even drug combinations to a wide variety of solid tumors.

## Methods

**Synthesis of STAT3 siRNA-loaded, iRGD albumin NP**. Albumin NPs were fabricated via the EHD jetting process previously established in our group. In brief, for all particle types, HSA was dissolved in a cosolvent mixture (80:20 v/v) of ultrapure water and ethylene glycol at a concentration of 7.5 w/v%. A bifunctional OEG (NHS-OEG-NHS, 2 kDa) was added at 10 w/w% relative to HSA. When iRGD was incorporated in the NPs, 355 ng per mg of albumin was added directly to the jetting solution. In contrast, when incorporating siRNA into the particles, the siRNA was first complexed with a branched polyethyleneimine (bPEI, 60 kDa) for 30 min in ultrapure water and the mixture was then added to the jetting solution resulting in 0.04 mg and 355 ng (26 pmol) of bPEI and siRNA per mg of protein NP, respectively. In instances when siRNA was not included (control NPs) an addition of bPEI in ultrapure water was still included. Final jetting solutions were pumped through a syringe equipped with a 26-gauge blunt tip needle at a flowrate of 0.1 mL h$^{-1}$ while a constant voltage (ranging from 7.5 to 9.0 kV) was applied to form a stable Taylor cone at the needle tip. Particles were collected in aluminum pans at a needle to collector distance of 15 cm and then incubated for seven days at 37 °C to facilitate complete polymerization. Albumin NPs were then stored in dark RT conditions in their dry state for future experiments.

**Collection and purification of albumin NPs**. Albumin NPs were collected according to a standard protocol developed in our lab. In brief, a small volume, 5–10 mL, of water:ethanol (80:20 v/v) + 0.5% Tween 20 was added to the aluminum pans containing EHD jetted NPs. The resulting NP suspension was gently sonicated to disperse any aggregates and passed through a 40 μm cell straining filter. The resulting solution was centrifuged at 4000 rpm (3220 × g) for 4 min to pellet and remove any albumin aggregates larger than 1 μm in diameter. The supernatant was then divided into 2 mL Eppendorf tubes and centrifuged at 15,000 rpm (21,500 × g) to concentrate the samples to a single 1 mL sample for use in planned experiments. Collected NPs were used within 1 week of their collection and were stored at 4 °C during that time.

**Characterization of albumin NPs size, shape, and concentration**. Albumin NPs were characterized prior to their use in any experiments to ensure they met specifications. Physical characterization included the measurement of particle size in both their dry and hydrated state. To measure particle diameter and investigate their morphology, small silicon wafers were placed on the grounded collection

surface and were subjected to the same incubation period to complete the step-growth polymerization. These samples were imaged via scanning electron microscopy (SEM) using a FEI NOVA 200 SEM/FIB instrument. Obtained SEM images were characterized using ImageJ software. NPs in their hydrated state were collected and purified as described above. The stock solution was diluted in PBS + 0.5% Tween 20 for subsequent measurements using DLS and NTA (nanoparticle tracking analyzer) to investigate size and solution concentration. DLS and NTA analysis was performed using the Malvern Nano ZSP and NanoSight NS300 instruments and software respectively. Albumin NP solution concentration was further validated using the BCA (bicinchoninic acid) assay.

**Albumin NP stability and swelling characterization**. Albumin NPs collected and purified as described above were studied to determine their swelling behavior in response to changes in pH and stability in their hydrated state. NPs from the concentrated stock solution were diluted into either a solution of PBS + 0.5% Tween 20 (pH 7.4) or sodium acetate – acetic acid buffer + 0.5% Tween 20 (pH 5.0). The final NP solutions were stored at 37 °C for a period of 10 days. Particle diameter was measured throughout this period to compare the particle size distribution in response to acidic vs. neutral pH conditions and their overall stability at physiological temperatures.

**Loading and release of siRNA from albumin NPs**. To validate siRNA loading and characterize its release from the albumin NPs, a fluorescently labeled, Silencer™ Cy3-labeled negative control siRNA was incorporated into the NP formulation following the same process as described above. NPs were incubated and collected as described above. To validate siRNA incorporation into the particles, a Alexa Fluor™ 488 BSA conjugate was added at 0.5% of the total albumin mass. Colocalization of the fluorescent albumin NP and siRNA were confirmed using super-resolution STED (stimulated emission depletion) microscopy. Imaging was performed with the University of Michigan Microscopy and Image Analysis Laboratory (MIL) Core's Leica 1× STED instrument and processed using the Leica LAS X software. The release of the same fluorescent siRNA was conducted over a period of four days in PBS + 0.5% Tween 20 at 37 °C. The supernatant was analyzed using a Horiba fluorimeter and compared against a previously generated calibration curve.

**Cell line and cell culture conditions**. GL26, GL26-Cit, and GL26-OVA GBM cells were cultured in Dulbecco's modified eagle (DMEM) media supplemented with 10% fetal bovine serum (FBS), 100 units mL$^{-1}$ penicillin, and 0.3 mg mL$^{-1}$ L-glutamine. For mCitrine and OVA selection, the culturing medium was additionally supplemented with 6 μg mL$^{-1}$ G418. Cells were passaged every 2–3 days and were maintained in a humidified incubator at 95% air/5% CO$_2$ at 37 °C.

**Immunofluorescence uptake/lysosome colocalization**. To study the effect iRGD has on particle fate upon uptake by glioma cells, GL26 cells were cultured as described above in 4-well Nunc™ Lab-Tek™ Chamber Slides. Cells were seeded at 50,000 cells per well and allowed to adhere overnight. Twelve hours after initial seeding, media was exchanged. Fresh media contained either iRGD albumin NPs or albumin NPs (without iRGD), each labeled by incorporating an Alexa Fluor™ 647 BSA, at a concentration of 13.3 μg NPs per mL. Thirty minutes after particle administration, the media was removed, cells were washed three times with warm PBS, and fresh NP-free media was then added to each well. Cells were cultured normally for an additional three hours before the cells were washed, fixed, and stained. In brief, cells were washed with PBS and fixed in 2% paraformaldehyde in PBS for 15 min and then permeabilized with 0.1% Triton X-100 in PBS for an additional 15 min. Cells were then rinsed three times with PBS, and five times with PBS + 0.5% BSA (PBB) and blocked with a one-hour incubation in 2% BSA in PBS. Following a rinse with fresh PBB, cells were incubated with primary antibody for LAMP-1, a lysosomal marker, at 5 μg mL$^{-1}$ in a PBB solution at room temperature for one hour. After five rinses with fresh PBB, a mixture of Goat Anti-Rabbit IgG H&L Alexa Fluor™ 555 and Phalloidin Alexa Fluor™ 488, prepared in PBB was

added and incubated for one hour at room temperature. Each well was rinsed five times with PBB, incubated for one minute with Hoescht at 0.01 mg mL$^{-1}$ in DI water, and washed three times with fresh PBS. Finally, the chamber portion of the device was removed, the glass slide allowed to dry and samples were mounted using Prolong™ Diamond Antifade Mountant to preserve the samples and protect against fluorescent signal bleaching. Once stained and mounted, each sample was imaged using the University of Michigan MIL Nikon A1SI confocal microscope and processed using NIS-Elements AR software. Settings for all samples were optimized and kept consistent from sample to sample. Z-stack images were collected from multiple regions within each well and the resulting three-dimensional images were analyzed using an established protocol and the CellProfiler software. Analyzed data were used to calculate the relative number of NPs internalized by the cells and the percent of these cells colocalized within the lysosomes.

**In vitro albumin SPNP delivered siRNA GFP silencing.** As a preliminary experiment, to validate the ability of siRNA delivered via the albumin NPs, particles loaded with siRNA against GFP were synthesized as described above. GL26-cit cells were cultured consistent with previously conducted experiments. Twelve hours after initial seeding at 50,000 cells/well in 4-well Nunc™ Lab-Tek™ Chamber Slides, NPs were administered at a concentration of $1.0 \times 10^{11}$ NP per mL. Cells were incubated with particles for a period of two hours before they were washed three times with PBS and fresh media was added to each well. Cells were then cultured for an additional five days. On each day, one sample from each experimental group was washed, fixed, and stained according to an established protocol. In brief, cells were washed three times with warm PBS and fixed in 3.7% paraformaldehyde solution in PBS for 15 min. Finally, cells were washed three times with fresh PBS, dried, and samples were mounted using Prolong™ Diamond with DAPI Antifade Mountant to both stain the nucleus and preserve the samples. Samples from each experimental group and time point were imaged using the University of Michigan MIL Nikon A1SI confocal microscope and processed with the NIS-Elements AR software. Multiple Z-stacks from each sample were taken, maintaining constant laser settings across all samples. GFP signal, normalized to that of the nucleus, was quantified using ImageJ software.

**In vitro STAT3 silencing via albumin SPNP delivered siRNA.** To validate the ability of STAT3 siRNA delivered via SPNPs, particles loaded with siRNA against STAT3 were synthesized as described above. Twelve hours after initial seeding of GL26 glioma cells at 300,000 cells/well in 6-well cell culture plates, NPs were administered at a concentration of $1.0 \times 10^{11}$ NP per mL. Cells were incubated with particles for a period of two hours before they were washed three times with PBS and fresh media was added to each well. Cells were then cultured for an additional three days with a daily exchange of fresh media. Whole-cell extracts were prepared by lysing the cells with RIPA buffer for 5 min on ice, then centrifuged at $10,000 \times g$ for 5 minutes at 4 °C to remove cellular debris. Protein concentration was quantified using the Pierce BCA Protein Assay Kit. STAT3 expression was quantified using the ProteinSimple capillary electrophoresis-based western blot assay and normalized to the expression of GAPDH. Relative protein expression was measured and analyzed using the Compass for SW software.

**Intracranial GBM models.** Six to eight-week-old female C57BL/6 were purchased from Jackson Laboratory (Bar Harbor, ME) and were housed in pathogen-free conditions at the University of Michigan. Animals were treated according to the University of Michigan Committee on Use and Care of Animals (IACUC) protocol PRO00007666. Immune-competent mice were housed in a pathogen-free, humidity (40%-60%) and temperature (65-75°F) controlled vivarium on a 12:12 h light:dark cycle (lights on 0700) with free access to food and water. Intracranial surgeries were performed in 6-8 week old C57BL/6 mice weighing 17–24 g in the University of Michigan Unit for Laboratory and Animal Medicine (ULAM). All experimental studies were performed in compliance with Institutional Animal Care & Use Committee (IACUC). Orthotopic tumors were established in C57BL/6 mice by stereotactically injecting 20,000 GL26, GL26-Cit or 60,000 GL26-OVA cells into the right striatum of the brain using a 22-gauge Hamilton syringe (1 μL over 1 min) with the following coordinates: +1.00 mm anterior, 2.5 mm lateral, and 3.00 mm deep.

**Intratumoral diffusion of iRGD-functionalized albumin SPNPs.** To assess Albumin NP accumulation within the GBM TME, Alexa Fluor™ 647 dye was loaded into albumin NP, which were administered intratumorally into GBM bearing mice. Fourteen days post GL26-mtomato tumor implantation, mice ($n = 2$/group) were intratumorally injected with $3.6 \times 10^8$ or $3.6 \times 10^9$ Alexa Fluor™ 647 loaded Albumin NP in 3 μL volume. From each group, mice were transcardially perfused 4 or 24 h after NP administration, and brains were processed for imaging. Alexa Fluor™ 647 dye loaded Albumin NP accumulation within the TME was imaged with confocal microscopy (Carl Zeiss: MIC System) at ×5 and ×20 magnification.

**Intravenous iRGD NP delivery.** To assess the accumulation of systemically administered iRGD-Albumin NPs within the GBM TME, Alexa Fluor™ 647 dye was loaded into the NPs, which were injected i.v. into GBM bearing mice. Seven days post GL26-cit tumor implantation, mice were i.v. injected with $2.0 \times 10^{11}$ NPs in 100-μL volume. From each group, mice were transcardially perfused at 4 h ($n = 3$

mice) and 24 h ($n = 3$ mice), and brains were processed for imaging. Accumulation of NPs within the TME was imaged with confocal microscopy (Carl Zeiss: MIC System) at ×63 with an oil-immersion lens.

**Biodistribution of iRGD NPs post systemic delivery.** To evaluate the biodistribution of iRGD-albumin NP in vivo, NPs were loaded with Alexa Fluor™ 647 dye. C57BL/6 mice bearing GL26 tumors ($n = 4$) were injected intravenously (i.v) with $2.0 \times 10^{11}$ Alexa Fluor™ 647 iRGD-albumin or albumin alone NPs in 100-μL volume 7, 10, and 13 days post tumor implantation. For the control group, tumor naïve mice were injected i.v. with $2.0 \times 10^{11}$ Alexa Fluor™ 647 iRGD-albumin or albumin alone NPs in 100-μL volume. From each group, mice were transcardially perfused 24 h post the last injection of NPs, and vital organs (i.e., lungs, spleen, liver, brain, and kidneys) were harvested. The fluorescent signal within each organ was measured with IVIS spectrum analysis.

To assess the iRGD-Albumin NPs' accumulation within the GBM TME, NPs loaded with Alexa Fluor™ 647 dye were administered i.v into GBM bearing mice. Seven days post GL26-cit tumor implantation, $2.0 \times 10^{11}$ NPs in 100-μL volume were administered i.v. to mice ($n = 4$). Then, mice were transcardially perfused at either 4 h ($n = 2$) or 24 h ($n = 2$) post the NP injection. Brains were collected and processed for imaging. Accumulation of NPs within the TME was imaged with confocal microscopy (Carl Zeiss: MIC System) at 63x with an oil-immersion lens.

**STAT3 expression following systemic administration of STAT3.** To validate the ability of STAT3 siRNA loaded SPNPs to reach the tumor in vivo, particles loaded with siRNA against STAT3 were synthesized as described above. C57BL/6 mice bearing GL26 tumors ($n = 5$) were injected intravenously (i.v) with $2.0 \times 10^{11}$ Alexa Fluor™ SPNPs, STAT3i SPNPs, or free STAT3i in 100 μL volume on 5, 7, 11, 15, 18, and 22 days post tumor implantation. For the control group, tumor-bearing mice were injected i.v with an equal volume of saline. Mice from each group were transcardially perfused with Tyrode's solution 24 h post the last injection of NPs or saline and brains were extracted. Tumors were dissected from the brain, and single-cell suspension was obtained. Whole-cell lysates were prepared by incubating the dissociated cells pellet with protease inhibitors and 1.4 mL RIPA lysis buffer on ice for 5 min. Resulting cell lysates were centrifuged at 13,000 rpm ($25,000 \times g$) at 4 °C for 10 min and supernatants were collected to determine protein concertation in comparison to standard bovine serum albumin (BSA) protein concentrations through bicinchoninic acid (BCA) assay. STAT3 and downstream target's protein expression was quantified using the ProteinSimple capillary electrophoresis-based western blot assay and normalized to total protein expression.

**Therapeutic study in tumor-bearing animals.** To evaluate the therapeutic efficacy of iRGD-Albumin NPs loaded with STAT3i, saline, $2.0 \times 10^{11}$ of empty SPNPs, STAT3i SPNPs or 330 μg of free STAT3i were administered intravenously in 100 μL volume to GL26 tumor-bearing mice on 5, 8, 11, 15, 18, 22, and 25 days post tumor implantation. Also, a dose of 2 Gy Irradiation (IR) was administered to tumor-bearing mice 5 days a week for two weeks at 7 days post tumor implantation. Each treatment group consisted of at least $n = 5$ mice. When mice displayed signs of neurological deficits, they were transcardially perfused with tyrodes solution and 4% paraformaldehyde (PFA).

**Blood cell counts and serum biochemistry.** Blood from GL26 GBM bearing mice was taken from the submandibular vein and transferred to EDTA coated microtainer tubes (BD Biosciences) or serum separation tubes (Biotang). Samples in the serum separation tubes were left at room temperature for 20 min to allow for blood coagulation before centrifugation at 2000 rpm ($400 \times g$). Complete blood cell counts and serum chemistry for each sample were determined by in vivo animal core at the University of Michigan.

**Liver histopathology.** Liver tissues from treated animals were collected following the completion of the full STAT3i SPNP + IR therapeutic regimen described in Fig. 4a. PFA-fixed tissues were embedded, sectioned at 4 μm, and stained with H&E. Histopathological characterization for each sample was performed by the in vivo animal core at the University of Michigan.

**Immunohistochemistry.** Using the vibratome system, PFA-fixed brains were serially sectioned 50-μm thick and placed in PBS with 0.01% sodium azide. IHC for macrophages was performed by permeabilizing the brain sections with TBS-0.5% Triton-X (TBS-Tx) for 5 min, followed by antigen retrieval at 96 °C with 10 mM sodium citrate (pH 6) for 20 min. Then, the sections were allowed to cool to room temperature (RT) and washed five times with TBS-Tx (5 min per wash). Next, the brain sections were blocked with 10% goat serum in TBS-Tx for 1 h at RT followed by overnight primary antibody F4/80 (BioRad, MCA497GA, 1:250) incubation at RT. The primary antibody was diluted in 1% goat serum in TBS-Tx. The next day, brain sections were washed with TBS-Tx 5 times and incubated in fluorescent-dye conjugated secondary antibody (Thermofisher, A21209, 1:1000) diluted in 1% goat serum TBS-Tx in the dark for 6 h. Finally, brain sections were washed in PBS 3 times and mounted onto microspore slides and coverslipped with ProLong Gold. High magnification images at ×63 were obtained using confocal microscopy (Carl Zeiss: MIC-System). H&E staining on

whole-brain sections was performed according to standard H&E staining protocols. Similarly, H&E staining was performed on livers that were embedded in paraffin and sectioned 5 μm thick using the microtome system. Brightfield images of the stains were obtained using Olympus BX53 microscope.

**ELISA.** To evaluate whether human albumin within the NPs is immunogenic, expression of antibodies against human albumin or mouse albumin in the serum of mice treated with saline, free NPs, free STAT3i, STAT3i SPNPs, IR, or STAT3i SPNPs + IR was determined using ELISA. In brief, 96-wells' plates were coated with NPs loaded with either mouse albumin or human albumin overnight at 4 °C. The coated plates were washed the next day with 1× PBS + 0.05% Tween 20 (PBS-Tween) five times, and then blocked with PBS-Tween containing 5% goat serum at RT for 2 h. This was followed by five more washes with PBS-Tween. Next, mouse serum diluted 1:100 from each treatment group was added to the NP coated wells in a 100 μL volume and incubated at 4 °C for 24 h. For a positive control, primary antibody against HSA (abcam, ab10241, 1:1000) or mouse serum albumin (abcam, ab34807, 1:1000) was added to the appropriate wells. The next day, the plates were washed five times with PBS-Tween and 100 μl of anti-mouse (Thermofisher, 62-6520, 1:3000) or anti-rabbit (abcam, ab6721, 1:3000) secondary antibody was added to the appropriate plate, followed by 1 h incubation at 37 °C. For a negative control, secondary anti-body against HSA or mouse serum albumin was added to the appropriate wells. Plates were then washed five times with PBS-Tween. Substrate solution (TMB) was added and the plate was incubated in the dark at RT for 30 min and the reaction was quenched by the addition of 2 M Sulfuric Acid. Plates were read on a 96-well plate reader (Spectramax Plus, Molecular Devices) at O.D. 450 nm.

**Flow cytometry.** Antibodies for flow cytometry analysis were obtained from Biolegend unless indicated otherwise. To assess the immune cell population within the GL26-OVA TME, two days post STAT3i alone, empty SPNP, STAT3i SPNP or STAT3i SPNP + IR treatment detailed above, mice were euthanized and the tumor mass with the brain was dissected and homogenized using Tenbroeck (Corning) homogenizer in DMEM media containing 10% FBS. Tumor-infiltrating immune cells were enriched with 30–70% Percoll (GE Lifesciences) density gradient and the cells were resuspended in PBS containing 2% FBS (flow buffer). Live/dead staining was carried out using fixable viability dye (eBioscience). Non-specific antibody binding was blocked with CD16/CD32. Dendritic cells were labeled with CD45, CD11c, and B220 antibodies. Plasmacytoid dendritic cells (pDCs) were identified as $CD45^+/CD11c^+/B220^+$ and conventional dendritic (cDCs) cells were identified as $CD45^+/CD11c^+/B220^-$. Macrophages were labeled with CD45, F4/80, and CD206 antibodies. M1 macrophages were identified as $CD45^+/F4/80^+/CD206^{low}$ and M2 macrophages were identified as $CD45^+/F4/80^+/CD206^{high}$. Tumor-specific T cells were labeled with CD45, CD3, CD8, and SIINFEKL-H2K$^b$-tetramer. Granzyme B and IFNγ were stained using BD intracellular staining kit following the manufacturer's instructions. For T cell functional analysis, purified immune cells from the TME were stimulated with 100 μg mL$^{-1}$ of GL26-OVA lysate for 16 h in DMEM media containing 10% FBS followed by 6-h incubation with Brefeldin and monensin. For integrin αvβ3 and αvβ5 analysis, untreated GL26-tumor-bearing mice were euthanized 23 DPI and both the tumor-bearing hemisphere and the contralateral hemisphere were dissected. Cells were dissociated from both the hemispheres into single-cell suspension and CD45 cells were labeled with magnetic beads (Miltenyi) using the manufactures' instructions at 4 °C. Purified cells were washed and passed through a preconditioned MS column placed in the magnetic field of a MACS separator. Cells that were negative for CD45 were collected, resuspended in flow buffer and labeled with αvβ3 (Novus, NBP2-67557) and αvβ5 (BD Bioscience, 565836) for flow cytometry analysis. All stains were carried out for 30 min at 4 °C with 3× flow buffer washes between live/dead staining, blocking, surface staining, cell fixation, intracellular staining and data measurement. All flow measurements have been made utilizing with FACSAria flow cytometer (BD Bioscience) and analyzed using Flow Jo version 10 (Treestar).

**Reporting summary.** Further information on research design is available in the Nature Research Reporting Summary linked to this article.

## Data availability

Data supporting the findings of this study are available within the article, its Supplementary Information files, and from the corresponding author upon reasonable request. A reporting summary for this article is available as a Supplementary Information file. Source data are provided with this paper.

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

## Acknowledgements

We thank S. Rahmani and L. Solario for training and guidance; L. Barthel and the BRCF Microscopy Core for help with confocal and STED imaging; and technical support from the Michigan Center for Materials Characterization for SEM imaging. This work has been supported by National Institutes of Health/National Institute of Neurological Disorders & Stroke (NIH/NINDS) Grants R37-NS094804, R01-NS074387, R21-NS091555 to M.G.C.; NIH/NINDS Grants R01-NS082311, and R01-NS096756 to P.R.L.; NIH/NINDS R01-EB022563; the Dr. Ralph and Marian Falk Medical Research Trust – Catalyst Awards Program, the Department of Neurosurgery; Leah's Happy Hearts, and ChadThough Foundation and Smiles for Sophie Forever Foundation to M.G.C. and P.R.L. RNA Biomedicine Grant F046166 and the Rogel Cancer Center Scholar Award to M.G.C.

## Author contributions

M.G.C. and J.L. proposed and supervised the project with input from E.R. and P.R.L.; J.V.G., P.K., and R.D. conducted experiments with assistance from M.C. and S.H.; J.V.G., P.K., M.G.C., and J.L. analyzed the data and wrote the manuscript with input from all co-authors.

## Competing interests

The University of Michigan has filed a patent application (US 62/931,512) on materials related to the work described in this manuscript. E.R. is a founder, shareholder and officer of DrugCendR, Inc., a company that is developing the iRGD peptide for clinical use.
