## [Peer Review File · Nature Communications]

Reviewers' Comments:

Reviewer #1:

Remarks to the Author:

This manuscript puts forward an interesting nanoparticle based RNAi delivery system which modifies the tumor immunological microenvironment in experimental glioblastoma multiforme (GBM). Nanoparticle based methods using synthetic materials to transport drugs and RNAi approaches across the blood brain barrier have largely been unsuccessful. The investigators have developed a new approach to construct a synthetic protein nanoparticle (SPNP) based on polymerized human serum albumin conjugated to the peptide iRGD which has established BBB penetrating properties. The payload for this SPNP is STAT3 silencing RNA (STAT3i) which is intended to polarize a more anti-glioma immune response in the brain. This team of investigators demonstrate STAT3i SPNPs: 1.) inhibit STAT3 in vitro, 2.) co-localize to brain macrophages upon intravenous injection, 3.) provide durable long term tumor eradication (when used in conjunction with irradiation), 4.) enhance CD8 T cell responses to model antigen ova, 5.) polarize macrophages to an M1 macrophage phenotype, 6.) protect against glioma rechallenge, and 7.) leave no obvious toxicities or overt tissue abnormalities. The supplemental data provided is appropriate and further enhances the manuscript. The data is compelling and appropriate statistical analysis is applied. The manuscript is also clearly written. In all, the findings outlined in this manuscript demonstrate an effective immune altering therapy which results in enhanced immunity to glioma.

In addition, there were some remaining questions that could be commented on:

1. The STAT3i SPNP clearly alters the GL26 microenvironment in the CNS. It is somewhat implied that this alteration occurs in the CNS at the level of APCs. However, given that the STAT3i SPNP is broadly distributed in multiple organs, is it possible this nanoparticle exerted its effects peripherally? The spleen is negative for STAT3i SPNP. However, it is possible the cLN could be involved?
2. The immune modulation of the STAT3i SPNP/irradiation combination therapy is profound. The analysis of augmented CD8 T cell responses is performed in the context of GL26 which express the model antigen ova. Does STAT3i SPNP induce alteration of CD8 T cell responses (or other immune cell types) in mice harboring gliomas that do not express ova?
3. The therapy is administered at 7 days post GL26 glioma inoculation. What is the size of GL26 gliomas at 7 days? Would this simulate an MRI detectable glioma in a GBM patient? Is there a limit at which point the combination therapy is no longer effective (i.e. day 14)?

Minor:

1. There appears to be a typo on the y-axis of Figure 3g – “IFB-y+”
2. Supplemental Figures 7 and 8 are referred to in the body of the paper.

Reviewer #2:

Remarks to the Author:

In this manuscript, Jason V. Gregory et al designed a synthetic protein nanoparticle (SPNP) equipped with the cell-penetrating peptide iRGD to deliver siRNA against STAT3 (STAT3i) to combat glioblastoma. They found that SPNP can efficiently deliver STAT3i to GBM in vitro and in vivo. In addition, when combined with the standard of care, SPNP exhibited antitumor efficiency in vivo and activated the immune system to obtain anti-GBM immunological memory. The topic is important and of great interest. However, the novelty of the paper is rather weak in term of the construction of the nanocarrier. In addition, there are some concerns remained to be addressed.

Major:

1. The novelty of the paper is rather weak in term of the construction of the nanocarrier. The

preparation method has been established by the same group, and the application of iRGD for GBM targeting is well known.

2. Most of the findings are descriptive, the mechanism how STAT3i-loaded SPNP combat GBM and how it prime an adaptive immune response should be addressed.

3. The application of high molecular weight branched PEI (60 KD) might cause severe cytotoxicity. The safety of the formulation to normal cells should be evaluated.

4. The preparation process of the iRGD-functionalized albumin nanoparticles were not clearly described. It is unclear how bPEI and iRGD are incorporated into the albumin nanoparticles. Can the authors describe the mechanism in more detail? A scheme can make the process easier to understand.

5. Intravenous injection route was applied for in vivo survival and therapeutic studies. However, intra-tumoral diffusion of iRGD-functionalized albumin nanoparticles was investigated in Supplementary Materials (page 7) and Fig.7. On the other hand, no results or discussions were mentioned in this manuscript. Please explain this.

6. In the section "Therapeutic study in tumor bearing animals" (page 10 line 4 in Supplementary Materials), various formulations were administered on 5, 8, 11, 15, 18, 22 and 25 days post tumor implantation. The injection interval was different. Why did authors choose this regimen?

Minor:

1. With respect to the measurements of the nanoparticle size, the authors should provide the polydispersity indices (PDI) corresponding to the measured sizes. Zeta potential data of the nanoparticles were missing in this manuscript.

2. In page 5 line 4, the authors claimed that "no significant change in particle size or morphology (Supplementary Fig. 2)". But in Supplementary Fig. 2 only particle size data was provided. Please provide the data about the particle morphology. The same problem also exists in page 6 line 22 "There were no significant differences in particle size, surface charge, or morphology between siRNA-loaded SPNPs and the control particles."

3. In page 7 line 8, based on the in vitro experiments, the authors chose 5 µg/kg as the dose to be used in vivo studies. How did authors calculate this dose based on the in vitro experiments results?

4. In page 13 line 3 and Fig.3f, "IFN-γ" was mis-written as "INF-γ". In addition, the explanation of abbreviation used should be added where it is first mentioned..

5. In page 2 line 5, "Signal Transducer and Activation of Transcription 3 factor (STAT3i) result in" should be revised as "Signal Transducer and Activation of Transcription 3 factor (STAT3i) resulted in". In page 2 line 7, "STAT3i SPNPs result in" should be revised as "STAT3i SPNPs resulted in". In page 2 line 8, "bearing mice and primes" should be revised as "bearing mice and primed".

Systemic Brain Tumor Delivery of Synthetic Protein Nanoparticles for Glioblastoma Therapy

Jason V. Gregory*, Padma Kadiyala*, Robert Doherty, Melissa Cadena, Samer Habel⁵ Erkki Ruoslahti, Pedro R. Lowenstein, Maria G. Castro[†], and Joerg Lahann[†]

Itemized Response to Reviewers:

Reviewer 1:

1. *The STAT3i SPNP clearly alters the GL26 microenvironment in the CNS. It is somewhat implied that this alteration occurs in the CNS at the level of APCs. However, given that the STAT3i SPNP is broadly distributed in multiple organs, is it possible this nanoparticle exerted its effects peripherally? The spleen is negative for STAT3i SPNP. However, it is possible the cLN could be involved?*

Answer: The reviewer raises an excellent point. In fact, we analyzed the maturation status of DCs in the draining lymph nodes (dLNs) of Free-STAT3i, SPNPs, STAT3i SPNPs and STAT3i SPNPs + IR treated GL26-OVA tumor bearing mice using flow cytometry. To examine the effect of SPNP on DC activation in the dLNs, the expression of MHC II was assessed. We observed an increase in the frequency of CD45⁺/CD11c⁺/MHC II⁺ (~1.5 fold, p<0.0001), DCs in the dLN of STAT3i SPNPs treated mice in compared to Free-STAT3i, SPNPs alone treated mice. This was further enhanced in the presence of IR by ~1.3 fold (p<0.0001) (please see **new Fig. S18, revised manuscript**). These data suggest that STAT3i SPNPs in combination with radiation induces the activation of DCs by enhancing the expression of MHC II, which is involved in antigen presentation.

2. *The immune modulation of the STAT3i SPNP/irradiation combination therapy is profound. The analysis of augmented CD8 T cell responses is performed in the context of GL26 which express the model antigen ova. Does STAT3i SPNP induce alteration of CD8 T cell responses (or other immune cell types) in mice harboring GBMs that do not express ova?*

Answer: We appreciate this insightful comment. The main reason why we used the GL26 model harboring the surrogate tumor antigen, ovalbumin (OVA) is that it enabled us to

assess the frequency of tumor antigen specific T cells within the GBM microenvironment, since OVA-specific MHC tetramers are available to determine this. We used this model to monitor antigen-specific T cells in the brain tumor microenvironment (TME). The induction of antigen-specific cytotoxic CD8⁺ T-cell responses were visualized using the OVA-derived peptide epitope SIINFEKL-H2K^b tetramer. In a GL26 model not harboring OVA, we will be able to identify the total number of T cells in the TME; however, antigen-specific T cells cannot be identified. To address the reviewer's comment directly, we demonstrated that STAT3i SPNPs enhance the median survival of mice bearing unmodified GL26 GBM (i.e., not expressing OVA) (Figure 3). Thus, we can conclude that the therapeutic benefits are not dependent on the expression of OVA. This explanation has now been included in the revised Discussion section, page 13, Line 17.

3. *The therapy is administered at 7 days post GL26 GBM inoculation. What is the size of GL26 GBMs at 7 days? Would this simulate an MRI detectable GBM in a GBM patient? Is there a limit at which point the combination therapy is no longer effective (i.e. day 14)?*

Answer: We appreciate this insightful comment and have now included histology of the tumor at 7 DPI as new Figure S10. The total tumor volume of this tumor is 9.61 mm³. However, we have not performed magnetic resonance studies in GL26 GBM bearing mice and can thus not conclude with certainty that the size of mouse GBM simulates an MRI detectable GBM corresponding to what would be observed in a GBM patient. We would need to utilize mathematical models with parameters set for diffusive as well as invasive growth patterns,¹ which is beyond the scope of our manuscript.

However, we have previously shown that GL26 murine model exhibits histopathological characteristics encountered in human GBM.² We have not assessed the impact of STAT3i SPNPs in combination with radiation on regression of larger tumors (i.e., day 14). Further studies will be required to explore the limit at which point our therapy will no longer be effective. We note, however, that in human patients, GBMs are resected maximally. Thus, patients would have minimal residual disease at the time of treatment with the STAT3i SPNPs. We have clarified the revised discussion section of our manuscript, page 8, Line 13 to reflect on these aspects.

Fig. S10. Volume of GL26 GBM at 7 DPI. C57BL/6 mice were implanted with 20,000 GL26 cells orthotopically and brains were processed for Nissl staining at 7dpi (scale bar = 1mm). Tumor volume =9.61mm³; tumor area =

4. *There appears to be a typo on the y-axis of Figure 3g – “IFB-y+”*

Answer: In the revised Figure 3g we changed ‘IFB-y+’ to IFN- γ +’.

5. *Supplemental Figures 7 and 8 are referred to in the body of the paper.*

Answer: In the revised manuscript, we have included additional discussions of supplemental Figures 7 (Now Supplementary Fig. 9) and 8 (Now Supplementary Fig. 11) in the results section, please see Page 7, line 12-23. And Page 10, lines 7-8.

Reviewer 2:

1. *The novelty of the paper is rather weak in term of the construction of the nanocarrier. The preparation method has been established by the same group, and the application of iRGD for GBM targeting is well known.*

Answer: We appreciate the comment made by the reviewer and agree that both, electrohydrodynamic co-jetting as well as the use of iRGD for GBM targeting have been established in previous publications. However, we note that there is still significant novelty in this paper in comparison to related publications: (i) *Novel Process*: This work, for the first time, extends electrohydrodynamic co-jetting to protein nanoparticles. In the past, this process has been exclusively used for synthetic polymers. (ii) *Novel Materials Design*: The protein nanoparticles have been uniquely designed for RNAi delivery, where PEI is simultaneously used as a linker to stabilize the protein particles (it reacts with the PEG macromer), and as a complexation agent for RNAi. (iii) *Novel use of tumor-homing peptides*: The iRGD is released locally from the synthetic nanoparticles which contrasts former approaches that either used systemic co-administration or surface-conjugated iRGD.^{3,4} The result is a synthetic protein nanoparticle technology that combines the versatility of engineered polymer particles with the biological characteristics of protein carriers.

In addition, an additional significant novel aspect of this paper relies on the biological effects elicited by this nanocarrier system. Specifically, we show that therapeutically relevant dose of siRNA can be delivered to tumor bearing animals to inhibit the expression of a key transcription factor, STAT3, which has previously been shown to mediate GBM progression. This treatment approach resulted in increased median survival of glioma bearing mice. When STAT3i-SPNP treatment was combined with radiotherapy, which is the standard of care for GBM, we observed 87.5% long term survivors. When the long-term survivors were rechallenged with GBM in the contralateral hemisphere, the animals did not succumb to tumor burden, indicating the development of anti-GBM immunological memory. Leading up to this study, published outcomes had been somewhat disappointing and studies that show some level of efficacy almost always rely on highly invasive convection-enhanced intracranial delivery – rather than the non-invasive, preferred systemic delivery route used in our study.

2. *Most of the findings are descriptive, the mechanism how STAT3i-loaded SPNP combat GBM and how it primes an adaptive immune response should be addressed.*

Answer: There is growing evidence that shows initial activation of a patient's immune system can result in tumor cell killing long after the completion of the treatment through the stimulation of the immune response against tumor antigens.⁵⁻⁷ Our data provide experimental evidence (Figures 3 and 4) that treating GL26 GBM bearing animals with STAT3i loaded SPNPs in combination with radiotherapy elicits an anti-tumor immune response by the recruitment of antigen presenting dendritic cells into the tumor mass, with concomitant induction of anti-GBM specific T cell clonal expansion, long-term survival, and immunological memory. In figure 3 we examined tumor-specific T cells in the TME of Free-STAT3i, SPNPs, STAT3i SPNPs and STAT3i SPNPs + IR treated GL26-OVA

tumor bearing mice. Tumor-specific T cells were identified using the SIINFEKL-H2K^b tetramer, the OVA cognate antigen (tumor antigen-specific T cells: CD3⁺/CD8⁺/SIINFEKL-H2K^b tetramer⁺). We observed a ~2.5 fold (p<0.0001) increase in tumor antigen-specific CD8⁺ T cells in the TME of mice treated with STAT3i SPNPs + IR compared other treatment groups. We also assessed the impact of STAT3i SPNPs on the activation status of CD3⁺/CD8⁺ T cells in the TME, by assessing their interferon- γ (IFN γ) and granzyme B (Gzb) expression levels. In CD8⁺ T cells isolated from the TME of mice treated with STAT3i SPNPs, IFN γ levels were ~4 fold higher (p<0.05) when compared to mice treated with Free-STAT3i, SPNPs. This response was further enhanced in the presence of radiation by ~2.4 fold (p<0.0001). Additionally, when the long term survivors from the STAT3i SPNPs + IR treatment group were rechallenged with GL26-wt tumors in the contralateral hemisphere (Figure 4), they remained tumor free without further treatment compared to control mice implanted with tumors, which succumbed due to tumor burden (MS: 28 days) (p<0.0001). These results indicate the development of immunological memory in the tumor bearing animals treated with STAT3i SPNPs + IR (Figure 4). Collectively, our data demonstrates that treatment of brain tumors with STAT3i loaded SPNP stimulates a systemic adaptive anti-GBM immune response, which could prevent GBM relapse.

In response to STAT3i SPNPs + IR, GBM cells undergo cell death with concomitant release of antigens and DAMPS into the TME,⁸ these antigens are picked up by infiltrating antigen presenting cells that migrate to the dLN where they would present antigens to T cells, leading to tumor antigen-specific T cell expansion and adaptive anti-GBM immunity. This has now been included in the revised discussion section, page 15, lines 1-9 and Fig. S18.

3. *The application of high molecular weight branched PEI (60 KD) might cause severe cytotoxicity. The safety of the formulation to normal cells should be evaluated.*

Answer: We appreciate this excellent comment by the reviewer. While we have not conducted a detailed toxicity study, we would like to offer several observations that suggest this particular formulation of protein nanoparticles does not cause severe cytotoxicity: (i) We investigated potential toxic effects of the protein nanoparticles to the liver of mice treated with protein nanoparticles. Supplementary Fig. 14 **shows histopathological analysis** confirming the absence of overt signs of liver toxicity - even after multiple nanoparticles administrations. (ii) We observed low levels of circulating antibodies against **human** serum albumin nanoparticles in mouse sera. In contrast, the circulating antibodies were undetectable, when human serum albumin was replaced with mouse serum albumin (Supplementary Fig. 15)-indicating that there is no immunogenicity for the serum albumin nanoparticles within the same species, i.e., mouse model and mouse albumin NP.

(iii) In our particle formulation, we are encapsulating the siRNA-PEI complex within the protein nanoparticle at very low concentrations. For comparison, Godbey et al. studied PEI-mediated gene delivery to EA.hy926 endothelial cells using high molecular weight (70 kDa) branched PEI⁹ and indeed found signs of cytotoxicity, especially at later time points, supporting the comment made by Reviewer 2. However, the PEI concentrations used in our study are 200-fold lower than what was used in the Godbey study. While we agree with reviewer that evaluation in cell culture might further confirm our results, based on the

Godbey et al study, we do not expect there to be toxicity in normal cells due to treatment with siRNA-PEI complex at the concentration that we use in this study. A more rigorous confirmation would require a larger cytotoxicity study in a relevant animal model and this is outside the scope of the current study.

4. *The preparation process of the iRGD-functionalized albumin nanoparticles was not clearly described. It is unclear how bPEI and iRGD are incorporated into the albumin nanoparticles. Can the authors describe the mechanism in more detail? A scheme can make the process easier to understand.*

Answer: Electrohydrodynamic (EHD) jetting uses a dilute aqueous solution of all components to be incorporated into the ultimate protein nanoparticle (here: HSA, PEG macromer, STA3i-PEI complex, iRGD). Once atomized, rapid evaporation of the solvent induces nearly instantaneous nanoprecipitation of all non-volatile components to form solid protein nanoparticles. A diagram showing the final protein nanoparticle development has been included in Supplementary Fig. 1. The PEG macromer covalently links the protein and PEI units resulting in a continuous network. The STAT3i is complexed to the PEI, and iRGD is encapsulated as detailed in Supplemental Fig. 1.

5. *Intravenous injection route was applied for in vivo survival and therapeutic studies. However, intra-tumoral diffusion of iRGD-functionalized albumin nanoparticles was investigated in Supplementary Materials (page 7) and Fig.7. On the other hand, no results or discussions were mentioned in this manuscript. Please explain this.*

Answer: GBM recurrence occurs locally, in regions adjacent to the tumor resection cavity. Local drug delivery at the time of surgery has the advantage of treating residual disease and preventing or prolonging the time to local recurrence. This is important because recurrence most often occurs locally after surgery.⁸ Therefore, we assessed whether iRGD-functionalized albumin nanoparticles intra-tumoral diffuse in the tumor bed (Supplementary Figure 9). Further, we observed that our NPs are effective at targeting GBM after intravenous injection (Figure 2), offering an attractive, non-invasive drug delivery platform to treat brain tumors. Thus, we utilized this drug delivery route to target STAT3 expressed in GBM tumors using STAT3i. In the revised manuscript we have discussed the results from supplementary figure 7 (Now Supplementary Fig. 9), please see page 7, lines 12-23.

6. *In the section “Therapeutic study in tumor bearing animals” (page 10 line 4 in Supplementary Materials), various formulations were administered on 5, 8, 11, 15, 18, 22 and 25 days post tumor implantation. The injection interval was different. Why did authors choose this regimen?*

Answer: The treatment regimen for both therapeutic efficacy study and phenotypic characterization of immune cellular infiltrates in response to STAT3i SPNP + IR in the GBM brain tumor microenvironment was identical. GBM bearing mice were intravenously injected with STAT3i SPNP on 5, 8, 11, 15, 18, 22 and 25 days post tumor implantation. We did not observe overt off target systemic toxicity in the livers of GBM bearing mice treated with this dosing regimen (Supplementary Figure 14) therefore we used this treatment schedule for in vivo studies.

7. *With respect to the measurements of the nanoparticle size, the authors should provide the polydispersity indices (PDI) corresponding to the measured sizes. Zeta potential data of the nanoparticles were missing in this manuscript.*

Answer: In the revised manuscript, both PDI and zeta potential data has been added for all particle types, please see Supplementary Figure 3 and Supplementary Figure 7.

8. *In page 5 line 4, the authors claimed that “no significant change in particle size or morphology (Supplementary Fig. 2)”. But in Supplementary Fig. 2 only particle size data was provided. Please provide the data about the particle morphology. The same problem also exists in page 6 line 22 “There were no significant differences in particle size, surface charge, or morphology between siRNA-loaded SPNPs and the control particles.*

Answer: In the revised manuscript, Supplementary Figure 2 (Now Supplementary Fig. 3) has been adjusted to include additional particle data. Additionally, Supplementary Figure 7, added in response to comment 7 above, supports the noted text on page 6.

9. *In page 7 line 8, based on the in vitro experiments, the authors chose 5 µg/kg as the dose to be used in vivo studies. How did authors calculate this dose based on the in vitro experiments results?*

Answer: Based on the average tumor volume at 7 DPI (~9.6 mm³) we estimated that the tumor contains approximately 2 million cells at the time which SPNP therapy is initiated. Using the previously demonstrated effective dose of 25 µg/mL of siRNA-loaded SPNPs referenced in our *in vitro* STAT3 knockdown experiments (Figure 1), along with our experimentally determined loading of 355 ng of siRNA per mg of SPNP, an approximate ratio of siRNA to tumor cells was determined. Despite targeting, it can be assumed that a very small fraction of administered particles reach the intracranial tumor site following systemic delivery, typically less than 1% of the injected dose. Together, this information was used to propose a dose of 5 µg/kg of siRNA for subsequent *in vivo* experiments. We recognize that challenges exist in relating *in vitro* experiments to *in vivo* dosing; however, our calculations were applied to arrive at a starting point from which adjustments could be made. While we found this calculated dose to be effective *in vivo*, plans were in place to either increase the administrated dose should we see minimal therapeutic effects or lower the dose if severe toxicity was observed.

10. *In page 13 line 3 and Fig.3f, “IFN-γ” was mis-written as “INF-γ”. In addition, the explanation of abbreviation used should be added where it is first mentioned.*

Answer: In page 13 line 22 and Fig.3f we corrected “INF- γ” to “IFN- γ.” Additionally we have added the explanation of the abbreviation in page 13 line 22 where it is first mentioned.

11. *In page 2 line 5, “Signal Transducer and Activation of Transcription 3 factor (STAT3i) result in” should be revised as “Signal Transducer and Activation of Transcription 3 factor (STAT3i) resulted in”. In page 2 line 7, “STAT3i SPNPs result in” should be revised as*

“STAT3i SPNPs resulted in”. In page 2 line 8, *“bearing mice and primes”* should be revised as *“bearing mice and primed”*.

Answer: In page 2 line 5 “Signal Transducer and Activation of Transcription 3 factor (STAT3i) result in” has been revised as “Signal Transducer and Activation of Transcription 3 factor (STAT3i) resulted in.” In page 2 line 7, “STAT3i SPNPs result in” has revised as “STAT3i SPNPs resulted in”. In page 2 line 8, “bearing mice and primes” has be revised as “bearing mice and primed”.

References

1. Rutter, E. M. *et al.* Mathematical Analysis of GBM Growth in a Murine Model. *Sci. Rep.* **7**, 1–16 (2017).
2. Candolfi, M. *et al.* Intracranial glioblastoma models in preclinical neuro-oncology: Neuropathological characterization and tumor progression. *J. Neurooncol.* **85**, 133–148 (2007).
3. Liu, X. *et al.* Targeted drug delivery using iRGD peptide for solid cancer treatment. *Mol. Syst. Des. Eng.* **2**, 370–379 (2017).
4. Sugahara, K. N., Teesalu, T., Karmali, P. P. & Ramana, V. Co-administration of a Tumor-Penetrating Peptide Enhances the Efficacy of Cancer Drugs. *Science (80-.)*. **328**, 1031–1035 (2010).
5. Curtin, J. F. *et al.* 22) HMGB1 mediates endogenous TLR2 activation and brain tumor regression. *PLoS Med.* **6**, e10 (2009).
6. Kamran, N. *et al.* Immunosuppressive Myeloid Cells’ Blockade in the GBM Microenvironment Enhances the Efficacy of Immune-Stimulatory Gene Therapy. *Mol. Ther.* **25**, 232–248 (2017).
7. Candolfi, M. *et al.* Temozolomide does not impair gene therapy-mediated antitumor immunity in syngeneic brain tumor models. **1**, 1–14 (2015).
8. Kadiyala, P. *et al.* High Density Lipoprotein-Mimicking Nanodiscs for Chemo-Immunotherapy against Glioblastoma Multiforme. **13**, 1365–1384 (2019).
9. Godbey, W. T., Wu, K. K. & Mikos, A. G. Poly(ethylenimine)-mediated gene delivery affects endothelial cell function and viability. *Biomaterials* **22**, 471–480 (2001).

Reviewers' Comments:

Reviewer #1:

Remarks to the Author:

The authors have adequately addressed my prior critique. I do not have additional concerns.

Reviewer #2:

Remarks to the Author:

The authors have addressed most of my concerns. However, as the nanoformulation mostly accumulated in the lung and liver (Fig 2d), toxicology data should be provided. The current discussion is too weak to support the safety of the nanoformulation.

Systemic Brain Tumor Delivery of Synthetic Protein Nanoparticles for Glioblastoma Therapy

Jason V. Gregory*, Padma Kadiyala*, Robert Doherty, Melissa Cadena, Samer Habel⁵ Erkki Ruoslahti, Pedro R. Lowenstein, Maria G. Castro[†], and Joerg Lahann[†]

Itemized Response to Reviewers:

Reviewer 2:

1. The authors have addressed most of my concerns. However, as the nanoformulation mostly accumulated in the lung and liver (Fig 2d), toxicology data should be provided. The current discussion is too weak to support the safety of the nanoformulation.

Answer: We appreciate Reviewer #2 comment. This reviewer raises an excellent point related to the safety of the nanoformulation. In response to this comment, we have now included toxicology data to answer this question (please see new Figure 4 and Supplementary Table 1). We assessed the complete blood cell counts (CBC), serum biochemistry for aminotransferase (ALT), bilirubin, urea (BUN) and creatinine for glioma bearing mice treated with saline, STAT3i, SPNPs, STAT3i-SPNP, and STAT3i-SPNP + IR. Our data show that CBC, circulating levels of ALT, BUN, and creatinine remain normal, indicating normal functioning liver and renal systems after STAT3i protein nanoparticles treatment in combination with radiation (**please see revised Figure 4 and Supplemental Table 1**). Histopathological analysis of the livers from the treatment groups detailed above confirm the absence of overt signs of liver toxicity - after multiple administration of STAT3i SPNPs, either alone or in combination with radiation. The mice were observed daily by a board certified veterinarian, and no overt signs of respiratory distress, adverse neurological signs or weight loss were observed. Collectively, these data strongly suggest that the proposed therapeutic combination does not induce systemic toxicity.

In the revised manuscript, we have now added this description to the Results Section (please see page 14, line 1) and further discussed the limitations of the current studies in relation to the safety of the nanoformulation (please see the revised discussion, page 15, line 10). Extensive toxicology studies will be implemented, when we submit our IND proposal to the FDA as a prelude to implementing this novel therapeutic strategy in a Phase I clinical trial for high grade gliomas. This extensive toxicology characterization will be an entire study on its own and is thus out of the scope of the present manuscript.

Reviewers' Comments:

Reviewer #2:

Remarks to the Author:

The authors have addressed my question.